# Scrupulosity in the Network of Obsessive-Compulsive Symptoms, Religious Struggles, and Self-Compassion: A Study in a Non-Clinical Sample

**Marcin Moroń [1,*](ID), Magdalena Biolik-Moroń [2] and Krzysztof Matuszewski [2]**

1   Institute of Psychology, University of Silesia, 40-007 Katowice, Poland
2   Institute of Theology, University of Silesia, 40-007 Katowice, Poland
*   Correspondence: marcin.moron@us.edu.pl

**Abstract:** Scrupulosity is a phenomenon of the intersection between religiosity and obsessive-compulsive disorder. It could be regarded as an interactive effect of religiosity, religious internal conflicts, cognitive distortions associated with thought processing and self-reference, and obsessiveness. The present study investigated scrupulosity in the network of religious/spiritual struggles, obsessive-compulsive disorder (OCD) symptoms, self-compassion, and religiosity in order to better describe a position of scrupulosity in the dimensions of mental health and illness. Two hundred and ninety-two religious individuals from Poland (two hundred and two women) between the ages of 18 and 83 (M = 39.3; SD = 13.7) participated in the study. We applied the Self-Compassion Scale, Religious and Spiritual Struggle Scale, Obsessive-Compulsive Inventory-Revised, Pennsylvania Inventory of Scrupulosity, and posed questions concerning identification with religious beliefs, the role of religion in one's identity, and religious attendance. Using correlation analysis and a network analysis, we demonstrated that scrupulosity was positively correlated with religious/spiritual struggles (mostly with moral struggles and religious doubts) and with obsessing as an OCD symptom. The bridge strength analysis indicated that scrupulosity may be regarded as a bridge symptom between religious/spiritual struggles and OCD symptoms. Pastoral and psychological counselling could use these results in order to design efficient treatments for people suffering from religious scruples.

**Keywords:** scrupulosity; religious/spiritual struggles; self-compassion; religiosity; obsessive-compulsive disorder

## 1. Introduction

Religion has been identified as one of the most common themes of obsessions (McKay et al. 2004) with prevalence among patients suffering from OCD, ranging from 0% to an even 93% (Foa et al. 1995; Greenberg and Huppert 2010; Huppert and Fradkin 2016; Siev et al. 2021). Among religious obsessions, the most frequently observed are recurrent fears that one has committed a sin, blasphemous thoughts of an intrusive nature, concerns about being not faithful or moral enough, fears that one didn't perform a religious prayer or ceremony properly, and worries about being punished by God (Abramowitz and Buchholtz 2020). Scrupulosity, which refers to the "persistent doubts about sin and irresistible urges to perform excessive religious behavior" (Abramowitz et al. 2002), is frequently recognized as a particular presentation of obsessive-compulsive disorder (OCD; Abramowitz and Jcoby 2014). Treatment results of OCD with scrupulosity symptoms are poor, which is attributed mainly to lack of knowledge about the nature of religious obsession among clinicians but also due to a reinforcement of clients' obsessions by members of their religious community (Huppert and Siev 2010). However, scrupulosity also appears with a different intensity in people without a clinical diagnosis of OCD (Abramowitz et al. 2002; Henderson et al. 2022). This indicates that scrupulosity could be regarded as a phenomenon

of the intersection between common religious/spiritual experiences and clinically-relevant obsessive thoughts.

Appearance of scrupulosity among non-patients is in line with basic propositions of cognitive-behavioral models of obsessions (Salkovskis et al. 1999), which state that clinically relevant obsessions may develop from common unwanted thoughts that are maladaptively regulated by an individual. Religion is a source of consolation and comfort for religious individuals, but also has the potential for strain and internal conflicts (Exline 2013). These internal conflicts, such as religious and spiritual struggle (Exline et al. 2014) or religious crises (Henderson et al. 2022) could be regarded as a source of unwanted thoughts which may develop into an obsession in particular individuals. Personal risk factors for development of clinical obsessions are an intolerance of uncertainty and other thought distortions (e.g., thought-action fusion; Abramowitz and Jcoby 2014), which are associated with poor insight, less self-compassion, and lower mindfulness (Deniz 2021; Leeuwerik et al. 2020).

The goal of the present study was to examine scrupulosity in the network of religious struggle, obsessive-compulsive symptoms, religious identification, and self-compassion. Based on the cognitive-behavioral model of scrupulosity (Abramowitz and Jcoby 2014), we investigated the associations between normal religious struggle, religiosity, and scrupulosity. Moreover, we investigated the links between self-compassion and scrupulosity, as self-compassion is correlated with cognitive distortions fostering conversion of unwanted thoughts into clinical obsessions (e.g., intolerance of uncertainty; Deniz 2021; Leeuwerik et al. 2020). Given the strong evidence that scrupulosity could in fact be a presentation of OCD, we also examined its associations with OCD symptomatology (Abramowitz et al. 2002). In the present study, we used a network analysis, which is a more contemporary approach to studying symptomatology in clinical psychology and psychiatry (Borsboom and Cramer 2013). A network analysis posits that mental disorders could be better understood and theorized as the result of a causal interplay between symptoms in a network structure (Borsboom 2017). According to the network approach, investigation of dynamic and mutual associations between symptoms could be clinically more relevant than an approach that was focused on latent processes that underlie the symptoms (Borsboom and Cramer 2013). Since scrupulosity is understudied in its position in the context of normative religious experiences and psychopathological symptoms (Siev et al. 2021), we used network analysis to exploratively analyze the position of scrupulosity in a network consisting of OCD symptomatology, religious and spiritual struggle, self-compassion, and religiosity. We also compared this approach with the usual approach based on latent variables in order to detect advantages and limitations of each approach in studying scrupulosity.

### 1.1. Scrupulosity

Scrupulosity is frequently referred to as "fearing sin where there is none" (Abramowitz and Buchholtz 2020). Clinical observations indicated approximately four presentations of scrupulosity: (a) ego-dystonic intrusive thoughts about sex, violence, immoral acts, etc. that are interpreted at least in part within a religious framework; (b) ego-dystonic thoughts specific to religion, e.g., images of holy figures or saints that would be generally considered blasphemous, accompanied by rituals and neutralizing strategies that may or may not involve religious themes; (c) ego syntonic thoughts of a religious nature, e.g., thoughts concerning questions of faith or interpretations of texts, which then develop into obsessions, in addition to checking and reassurance-seeking rituals, and (d) obsessional doubts about whether religious rules and commandments have been followed correctly, or whether one is "faithful enough" (Abramowitz and Jcoby 2014, p. 141). Two dimensions of scrupulosity identified in previous studies using the Penn Inventory of Scrupulosity (PIOS; Abramowitz et al. 2002)—the only psychometrically validated self-report measure of scrupulosity available to date (Abramowitz and Jcoby 2014)—on clinical and non-clinical samples are: (a) the fear of having committed a religious or moral sin (Fear of sin), and (b) the fear of punishment from God (Fear of God; Abramowitz et al. 2002; Olatunji et al. 2007).

Recent validation of the PIOS indicated that it consists of two factors which are: (a) Fear of God and (b) Fear of immorality (Huppert and Fradkin 2016). Clinically relevant symptoms of scrupulosity captured by the PIOS include a strong fear of God's punishment and of a lack of God's acceptance, strong preoccupation with avoiding immoral thoughts, frequent fears of having immoral sexual thoughts, a fear of acting immorally without being aware of it, and strong feelings of guilt (Huppert and Fradkin 2016). In the current operationalization, scrupulosity is treated as a dimension ranging from less severe fears of being immoral to an extreme, obsessive fear of being immoral (Abramowitz et al. 2002). OCD patients suffering clinically relevant religious scruples reported levels of scrupulosity measured more highly by the PIOS compared to individuals reporting other OCD-presentation and other diagnoses (e.g., anxiety disorders; Huppert and Fradkin 2016; Siev et al. 2011).

According to the general cognitive-behavioral approach to obsessional problems (Salkovskis et al. 1999), unwanted and intrusive thoughts that are contrary to one's moral or religious belief system are normal for almost everyone from time to time (Abramowitz and Jcoby 2014). Thus, religiosity and occasional unwanted thoughts associated with religion could not be treated as an antecedent of clinically relevant religious obsessions (Abramowitz and Buchholtz 2020). However, if unwanted religious thoughts are accompanied with beliefs about the importance of those thoughts and an intolerance of uncertainty (e.g., thought-action fusion; Shafran et al. 1996), they may develop into clinical obsessions (Abramowitz and Jcoby 2014). This indicates that scrupulosity could constitute a dimension origin in common religious struggle or doubts (Henderson et al. 2022), which may develop into a pathological presentation as OCD disorder in the presence of risk factors associated with poor insight and avoiding uncertainty (Tolin et al. 2001).

*1.2. Religious/Spiritual Struggles and Scrupulosity*

A number of studies have shown positive associations between religiosity and scrupulosity (Abramowitz and Buchholtz 2020; Henderson et al. 2022). People who were more religious reported more scrupulosity. Lau and Ramsay (2019) demonstrated that among religious individuals, scrupulosity is responsible for poorer well-being. However, being religious is not causally related to scrupulosity (Abramowitz and Buchholtz 2020). Moreover, few studies have inquired directly into religious/spiritual doubts or struggles, concerns which could be regarded as natural precursors to religious obsessions. Recently, religious crisis has been shown to be positively correlated with scrupulosity and thought-action fusion, which is regarded as a triggering factor in clinical obsession development (Henderson et al. 2022).

A construct which refers to tension, strain, and conflicts about sacred matters is a religious struggle (Exline et al. 2014). Its associations with scrupulosity were not extensively examined in this study. Several forms of religious and spiritual struggles were distinguished. Divine struggle regards as negative some emotions centered on beliefs about God, or a relationship with God (e.g., being angry at God). Demonic struggles involve beliefs that the devil or evil spirits are attacking an individual. Interpersonal struggles refer to concerns about negative experiences with people or institutions representing one's religion. Moral struggle means wrestling with attempts to follow moral rules of a religion and encompasses feelings of guilt about perceived transgressions committed. Ultimate meaning struggle involves concerns about a deeper purpose of life. Religious doubt involves uncertainty about religious truths (Exline et al. 2014; Zarzycka et al. 2020). Meta-analytical studies have demonstrated that religious struggles have been positively associated with poor mental health (Ano and Vasconcelles 2005; Smith et al. 2003).

Given that scrupulosity includes intrusive thoughts about being immoral or sinful, individuals experiencing frequent moral or demonic religious struggles could be more at risk of developing clinical obsessions concerning these religious themes. Strong divine struggles could also be regarded as sinful, mainly when accompanied with strong thought-action fusion (Abramowitz and Jcoby 2014). Interpersonal struggles, which tend to regard or perceive other members of a religious community as wrongful, may develop through concerns

about one's inability to love others despite their deeds, which is a tenet of teachings in some religious organizations, such as the Roman Catholic Church. Thus, in the present study we hypothesized that scrupulosity could be positively associated with religious struggle.

### 1.3. Self-Compassion and Scrupulosity

Individuals suffering religious obsessions have poor insight, experience more perceptual distortions, and have higher magical ideation compared to those with other types of obsessions (Tolin et al. 2001). They also tend to have a higher level of thought-action fusion and an intolerance of uncertainty (Abramowitz and Buchholtz 2020). An important construct that is associated with fewer distortions than are present in scrupulosity presentation of OCD is self-compassion (Neff 2003a, 2003b). Neff (2003b) defines self-compassion as "being touched by and open to one's own suffering, not avoiding or disconnecting from it, generating the desire to alleviate one's suffering and to heal oneself with kindness. Self-compassion also involves offering nonjudgmental understanding to one's pain, inadequacies and failures, so that one's experience is seen as part of the larger human experience." In this vein, self-compassion reflects a healthy attitude and relationship with oneself, which is in line with meta-analysis indicating positive associations between self-compassion and mental health (MacBeth and Gumley 2012).

Self-compassion encompasses three components and six dimensions (Neff 2003b). The first component is self-kindness, which is understood as gentleness and understanding toward oneself. The second dimension of this component is self-judgement, which refers to a sharp criticism toward oneself. The second component is common humanity, namely the individual's conviction that bad or difficult things happen not only to them but are characteristic of most people's experiences. The second dimension of this component is isolation. The third component is mindfulness, and consists of continuing to be aware of one's experience. The second dimension of this component is over-identification, which refers to exaggerating or ignoring specific aspects of experience, such as the pain one is feeling (Kocur et al. 2022).

Previous studies have found that scrupulosity was correlated positively with thought–action fusion (Siev et al. 2017a) and intolerance of uncertainty (Nelson et al. 2006). Self-compassion and mindfulness were associated with a lower intolerance of uncertainty (Mantzios et al. 2015). Mindfulness-based interventions were effective in reduction of thought-action fusion and intolerance of uncertainty among OCD patients (Asli Azad et al. 2019). Thus, self-compassionate and mindful individuals might avoid the development of religious scruples due to their higher tolerance of uncertainty and lower thought-action fusion. Self-compassion was also correlated with lesser maladaptive perfectionism (Stoeber et al. 2019). Among religious individuals, self-compassion was associated with less perfectionism and with a perception of greater support and forgiveness received from God (Brodar et al. 2015). Scrupulosity represents a strong fear of immorality (Huppert and Fradkin 2016) that could be regarded as a reflection of religious perfectionism, which is lower among self-compassionate individuals (Brodar et al. 2015). Research studies on the direct association between mindfulness and scrupulosity are rare. However, they indicate that mindful individuals experience fewer religious scruples. Mindfulness, and particularly, nonjudging, was negatively correlated with scrupulosity in a sample of undergraduate students (Fisak et al. 2019). Self-compassion was negatively associated with scrupulosity among men (Borgogna et al. 2020). These results suggest that scrupulosity develops easily among less mindful and more self-judging individuals.

### 1.4. Obsessive-Compulsive Symptoms and Scrupulosity

Religious individuals with OCD are more likely than nonreligious individuals with OCD to have symptoms of a religious nature. However, religious individuals are not, as a whole, more likely to have OCD (Siev et al. 2017b). Due to suggestions that scrupulosity could be a presentation of OCD, or even a separate form of OCD disorder (Abramowitz

and Jcoby 2014), scrupulosity was frequently tested with regard to OCD symptomatology in both clinical and non-clinical samples.

In the commonly-used approach to studying obsessive-compulsive symptomatology (Obsessive-Compulsive Inventory; Foa et al. 2002), several types of obsessions were distinguished, including: (a) washing (e.g., washing or cleaning oneself because of feeling contaminated), (b) checking (e.g., repeatedly checking gas and water taps), (c) ordering (e.g., getting upset if objects are not arranged properly), (d) hoarding (i.e., collecting things without a specific need), (e) obsessing (i.e., difficulties in controlling one's own thoughts), and (f) neutralizing (e.g., a need to repeat certain numbers). Early studies demonstrated positive associations with many dimensions of OCD symptomatology (e.g., washing, checking, doubting, slowness; Abramowitz et al. 2002). Later studies showed the strongest associations with obsessing (Nelson et al. 2006). Thus, in the present study, we expected that scrupulosity would correlate positively mainly with obsessiveness, while its associations with other dimensions of OCD symptoms would be weaker.

*1.5. A Network Analysis Approach*

The traditional conceptualizations of psychopathology presumed that symptoms of mental problems were reflective of underlying diseases ("latent" common causes; Cramer et al. 2010). In the network approach, symptoms are conceptualized as elements of a complex dynamical system, one in which they interact with each other (Borsboom and Cramer 2013). The mutual associations between symptoms rather than the latent structure of the symptoms are investigated, using a graphical depiction as a network of associated symptoms. Symptoms can be activated by other symptoms in the network (e.g., demonic struggle can activate fears of sin and being obsessed). This refers to the common clinical knowledge that symptoms can reinforce one another, leading to symptom cycles without any underlying latent process (Cramer et al. 2010). Specifically, a network structure of symptoms consists of "nodes" referring to the selected variables indicating the symptoms, and "edges" representing the associations that connect the nodes (e.g., regularized partial correlation coefficients). Network analysis helps also to investigate the centrality of symptoms and their communities, not necessarily assuming that latent processes are responsible for the detected associations between symptoms (Borsboom and Cramer 2013).

In the study of scrupulosity, this approach seems particularly important. Mental health professionals, and also patients, have problems with differentiating between the normative religious struggle and pathological symptoms of scrupulosity. Religious doubts and rituals may appear to be similar to compulsive rituals and obsessions present in a mental disorder (e.g., OCD; Siev et al. 2021). The question of how religious individuals become scrupulous is similarly under-studied (Siev et al. 2017b). People who are highly religious could develop scruples due to their rigid religiosity (Henderson et al. 2022), but also due to their particular self-reference which lacks self-compassion and is overly perfectionistic (Brodar et al. 2015). Investigating the associations between symptoms of normative religious experiences, scrupulosity, OCD symptoms, and self-reference could help in determining which symptoms are responsible for the potential transformation of normative religious doubts into clinically relevant obsessions, i.e., by determining which of them are bridge symptoms between normative processes and clinically-relevant symptoms. (See a similar approach to parental burnout in Blanchard and Heeren 2020).

*1.6. The Current Study*

The goal of the current study is to investigate scrupulosity in the network of (a) common religious and spiritual struggles (Exline et al. 2014), which represent a potential source of unwanted thought for religious people; (b) OCD symptomatology, which could be associated with obsessive religious scruples; (c) religiosity, represented by positive identification with one's religion; and (d) self-compassion, which reflects a healthy attitude toward oneself and an ability to tolerate internal uncertainty and conflicts (Neff 2003a). We expected to find a positive association with religious struggle in the obsessing symptoms of

OCD, and self-judgement, isolation, and over-identification as signs of low self-compassion. Contrarily, we expected a negative association between self-kindness, common humanity, and mindfulness with regard to scrupulosity. According to previous studies, we expected a positive association between scrupulosity and religiosity (Abramowitz and Buchholtz 2020). Since scrupulosity was not frequently examined in the context of dynamic associations between religious struggles, psychopathological symptoms and self-compassion, we used the network analysis in the present study. This approach helped to detect central symptoms of the network and bridge symptoms which could be responsible for the activation of other symptoms in the network, and to extract the communities of symptoms (namely, clusters of symptoms which are strongly interconnected but less correlated with other symptoms present in the network. However, we also used the conventional structural equation modeling (SEM) in order to examine the associations between self-compassion, lack of self-compassion, religiosity, religious struggles, OCD symptoms, and scrupulosity.

## 2. Materials and Methods

The present study employs a cross-sectional design based on the quantitative measurement of religious struggles, scrupulosity, self-compassion, and OCD symptoms, controlling for basic aspects of religiosity. This approach allows a reliable and valid assessment of the intensity of scrupulosity and its associations with other variables. Thus, we use the Pennsylvania Inventory of Scrupulosity, which is currently the most significant measure of assessment of religious scrupulosity, one particularly suitable in discriminating scrupulous obsessions in Christians (Huppert and Fradkin 2016).

### 2.1. Participants and Procedure

Inclusion criteria for participants in the study were: (a) being over the age of 18; (b) describing oneself as a religious individual; and (c) because the study was conducted in Poland, the participants should be Polish native speakers. The sole exclusion criterion was a clinical diagnosis of obsessive-compulsive disorder. Three hundred sixteen individuals participated in the present study via on-line questionnaire. Nine persons were excluded from participation due to describing themselves as non-religious (2.85%) and 18 individuals reported having a clinical diagnosis of OCD (5.70%). After exclusion of individuals who did not meet criteria, the remaining group of eligible participants consisted of 292 individuals (202 women; 69.18%). The age of the participants ranged from 18 to 83 years (M = 39.3; SD = 13.7). One hundred and eighty-five participants reported higher education (63.4%), seventy-nine reported secondary education (27.0%), fourteen reported post-secondary education (4.8%), ten individuals reported vocational education (3.4%) and four people reported primary education (1.4%). All of the participants were Roman Catholics.

The sample size was determined based on the recommended sample size for obtaining stable correlation estimates (*N* = 250; Schönbrodt and Perugini 2013). Thus, the number of participants in the present study met this criterion. The study was approved by the institutional ethics board (KEUS 244/04.2022). The participants were invited to complete on-line questionnaires through an invitation posted on social media. Their participation was voluntary, without any compensation. After receiving a description of the study, the participants indicated that they agreed with the study conditions and gave informed consent. They were afterward presented with the measures used in the present study and thanked for their participation.

### 2.2. Measures

The following measures were used:

- The Self-Compassion Scale (Neff 2003a; Kocur et al. 2022) consists of 26 items measuring six aspects of self-compassion: (a) self-kindness (e.g., "I try to be loving toward myself when I'm feeling emotional pain"; 5 items), (b) self-judgement (e.g., "When times are really difficult, I tend to be tough on myself"; 5 items), (c) common humanity (e.g., "When things are going badly for me, I see the difficulties as part of life that

everyone goes through"; 4 items), (d) isolation (e.g., "When I'm feeling down, I tend to feel like most other people are probably happier than I am"; 4 items), (e) mindfulness (e.g., "When something upsets me I try to keep my emotions in balance"; 4 items), and (f) over-identification (e.g., "When I'm feeling down I tend to obsess and fixate on everything that's wrong"; 4 items). Items were assessed on scale ranged from 1 (*Almost never*) to 5 (*Almost always*). Higher scores on the self-kindness, common humanity and mindfulness items indicate higher self-compassion. Higher scores on the self-judgement, isolation and over-identification items indicate less self-compassion. The scale has been found to be reliable and valid in previous Polish studies (Kocur et al. 2022).

- The Obsessive-Compulsive Inventory—Revised (Foa et al. 2002; Polish version: Jeśka 2012) consists of 18 items measuring six OCD symptoms: (a) washing (e.g., "I wash my hands more often and longer than necessary"; 3 items), (b) obsessing (e.g., "I find it difficult to control my own thoughts"; 3 items); (c) hoarding (e.g., "I collect things I don't need"; 3 items); (d) ordering (e.g., "I get upset if objects are not arranged properly"; 3 items), (e) checking (e.g., "I check things more often than necessary"), and (f) neutralizing (e.g., "I feel I have to repeat certain numbers"; 3 items). Participants assess the items on the scale ranging from 0 (*Not at all*) to 4 (*Extremely*). The scale has been found to be reliable and valid in previous studies with Polish participants (Brytek-Matera et al. 2022).

- The Religious and Spiritual Struggles Scale (RSS-14; Exline et al. 2022) is a version of the RSS (Exline et al. 2014) abbreviated in order to make it a more useful tool for research and practice. In the present study, we used Polish wordings of items from the full scale (Zarzycka et al. 2018) and created a selection for the abbreviated version (Exline et al. 2022). The RSS-14 measures six types of spiritual and religious struggles: (a) divine (e.g., "felt as though God had abandoned me"; 3 items), (b) demonic (e.g., "felt attacked by the devil or by evil spirits"; 2 items), (c) interpersonal (e.g., "had conflicts with other people about religious/spiritual matters"; 3 items), (d) moral (e.g., "wrestled with attempts to follow my moral principles"; 2 items), (e) doubts (e.g., "felt troubled by doubts or questions about religion or spirituality"; 2 items) and (f) ultimate meaning (e.g., "questioned whether life really matters"; 2 items). In the present study we did not use the ultimate meaning struggle scale due to a lower similarity of its content with religiosity. Participants assessed how frequently in the past month they had experienced a particular struggle using the scale from 1 (*Not at all*) to 5 (*A great deal*).

- The Pennsylvania Inventory of Scrupulosity (PIOS; Abramowitz et al. 2002) consists of 19 items and measures two dimensions of scrupulosity: (a) Fear of sin (e.g., "I feel guilty about immoral thoughts I have had"; 12 items) and (b) Fear of God (e.g., "I am afraid my behavior is unacceptable to God"; 7 items). Participants rate each item on the scale ranging from 0 (*Never*) to 4 (*Constantly*). Although the PIOS has had some revised versions (e.g., Olatunji et al. 2007), we decided to translate the original version. Thus, we used a back-translation procedure with three Polish psychologists fluent in English who translated the original items into Polish, and one professional English editor who translated a unified Polish version back to English. The internal structure of the PIOS was initially tested with confirmatory factor analysis (CFA). The two correlated factors model fit the data well when four covariances between items were added due to modification indices inspection ($\chi^2 = 428.431$; $p < 0.001$; CFI = 0.933; TLI = 0.922; RMSEA = 0.081; SRMR = 0.043). Our approach, based on covariances addition after modification indices inspection, was similar to the procedure used by Olatunji et al. (2007). However, future studies should investigate the internal structure of the PIOS more in-depth. In the present study, we are focused on the position of symptoms of scrupulosity in the network, thus an investigation of the latent structure of the scrupulosity itself was not a merit of the study. According to previous studies,

scores above 1.42 for the total score of the PIOS represent clinically relevant levels of scrupulosity as the manifestation of OCD.

- Religiosity was measured with three items: "I identify strongly with my religious beliefs", "My religion is an important part of my identity", and "How frequently do you attend religious ceremonies?" The first two items were assessed on the scale ranged from 1 (*Completely disagree*) to 5 (*Completely agree*). They were based on the items used by Siev et al. (2011), which concern the importance of religious beliefs and the role of religion in one's identity. The last item was assessed on the scale from 0 (*Never*) to 4 (*Very frequent*). The last item is frequently used as an indicator of religiosity (Abramowitz et al. 2002; Fincham and May 2019). These three items have high internal consistency ($\alpha$ = 0.902) and were indicators of a single latent variable ($\chi^2$ = 0; RMSEA = 0; SRMR < 0.001; CFI = 1.00; GFI = 1.00).

### 2.3. Statistical Analysis

First, descriptive statistics (means, standard deviations, and internal consistency using Cronbach's alpha) were calculated for all study variables. Second, we used a correlational analysis to inspect the associations between religious/spiritual struggle, obsessive-compulsive symptoms, self-compassion, religiosity, and scrupulosity. Due to the large number of correlation coefficients estimated in the study, we corrected the alpha level to $\alpha$ = 0.001 in order to detect significant correlation coefficients.

Next, we estimated a network model using all study variables. In the weighted network estimated in the present study, "nodes" represent the studied variables while "edges" (links connecting two nodes) represent the regularized partial correlation coefficients (controlled for all other nodes). The regularized partial correlation coefficient networks were estimated using the Network module implemented in JASP 0.14.1.0. We applied the least absolute shrinkage and selection operator (LASSO) with a tuning parameter selected by minimizing the Extended Bayesian Information Criteria (EBIC; using the default value of hyperparameter $\gamma$ = 0.5) (Epskamp et al. 2018).

In order to estimate the network accuracy, we used three steps: (A) estimation of the accuracy of edge-weights using bootstrapped confidence intervals (CI); (B) investigation of the stability of (the order of) centrality indices; and (C) investigation of differences between edge-weights and centrality indices using bootstrapping methods (Epskamp et al. 2018).

First, we examined each node using three indices of node centrality (Opsahl et al. 2010): node strength, betweenness, and closeness. Node *strength* refers to the number and strength of the direct connections of a node. *Betweenness* is a measure of how often a node lies on the shortest path between every combination of two other nodes. Thus, betweenness indicates to what extent the node facilitates the flow of information through the network. *Closeness* refers to the average distance from a node to all other nodes in the network, representing how fast a node can be reached from them. *Expected influence* refers to the sum of a node's connections and reflects the relative importance of a node in a network (Robinaugh et al. 2016).

We also examined edge-weights stability by estimating confidence intervals (CI). To estimate the stability of node centrality, we used the central stability coefficient (*CS*-coefficient) representing the proportion of participants that can be dropped from the analysis, such that the correlation between the original centrality indices and the subset centrality indices is at least 0.7 with a 95% probability (Beard et al. 2016; Epskamp et al. 2018). Additionally, we examined bridge strength and bridge expected influence in order to investigate which symptoms of a given psychological construct have the strongest associations with the symptoms of the other construct (Kaiser et al. 2021). Finally, we investigated communities of symptoms using the spin glass algorithm, which is a modularity-based community detection procedure suitable for uncovering the structure of networks (Blanchard et al. 2021). We used the spinglass.community function ($\gamma$ = 1, start temperature = 1, stop temperature = 0.01, cooling factor = 0.99, spins = 7) of the R package igraph. If the detected community consists of symptoms of different concepts, the structural equation modeling

(SEM) will be used to determine whether one latent variable underlies symptoms included in the community. In accordance with the recommendations, the good model fit was defined by the following criteria (Hu and Bentler 1999): RMSEA < 0.06; CFI > 0.95; TLI > 0.95; and SRMR < 0.08.

SEM was also used to examine the latent variables represented by symptoms of self-compassion, lack of self-compassion, religious struggles, religiosity, and OCD symptomatology as predictors of scrupulosity. In the structural model, we assumed covariations between the predictor latent variables.

## 3. Results

### 3.1. Descriptive Statistics

The means, standard deviations, distributions, and reliability of all studied variables are given in Table 1. The mean total score of scrupulosity in the sample was lower than the cut-off for the clinically relevant level of scrupulosity (Huppert and Fradkin 2016). However, a violin boxplot (Figure 1) shows that 36% of the participants reported scores higher than this cut-off.

**Table 1.** Descriptive statistics for the study variables.

| Variable | M | SD | Skewness | Kurtosis | α |
|---|---|---|---|---|---|
| Self-Compassion | | | | | |
| Self-Kindness | 3.145 | 0.832 | −0.061 | −0.280 | 0.847 |
| Self-Judgment | 3.014 | 0.801 | −0.204 | 0.151 | 0.809 |
| Common Humanity | 3.040 | 0.707 | −0.158 | 0.045 | 0.731 |
| Isolation | 3.149 | 0.886 | −0.127 | −0.431 | 0.785 |
| Mindfulness | 3.222 | 0.712 | −0.059 | −0.211 | 0.726 |
| Over-Identification | 3.144 | 0.825 | −0.168 | −0.299 | 0.747 |
| Obsessive-Compulsive Symptoms | | | | | |
| Washing | 0.911 | 0.882 | 0.736 | −0.417 | 0.669 |
| Obsessing | 1.211 | 0.961 | 0.556 | −0.438 | 0.787 |
| Hoarding | 1.659 | 0.896 | 0.289 | −0.457 | 0.586 |
| Ordering | 1.555 | 0.965 | 0.321 | −0.432 | 0.723 |
| Checking | 1.495 | 1.024 | 0.440 | −0.461 | 0.757 |
| Neutralizing | 0.689 | 0.900 | 1.199 | 0.373 | 0.810 |
| Religious/Spiritual Struggles | | | | | |
| Divine | 1.928 | 0.988 | 0.968 | 0.338 | 0.850 |
| Demonic | 1.753 | 0.993 | 1.100 | 0.376 | 0.758 A |
| Interpersonal | 2.183 | 1.003 | 0.542 | −0.368 | 0.753 |
| Moral | 2.688 | 1.047 | −0.008 | −0.856 | 0.592 A |
| Doubt | 2.265 | 1.211 | 0.648 | −0.420 | 0.771 A |
| Religiosity | | | | | |
| Religious identification | 3.870 | 1.073 | −0.646 | −0.317 | - |
| Religious identity | 4.092 | 1.043 | −1.010 | 0.400 | - |
| Religious attendance | 3.705 | 1.082 | −0.523 | −0.766 | - |
| Scrupulosity | 1.171 | 0.822 | 0.558 | −0.276 | 0.960 |
| Fear of Sin | 1.151 | 0.819 | 0.492 | −0.408 | 0.937 |
| Fear of God | 1.206 | 0.901 | 0.637 | −0.212 | 0.915 |

Note, A—correlation between two items.

Correlation coefficients are given in Table 2. Scrupulosity was positively correlated with Obsessive-Compulsive Symptoms; the correlation coefficients ranged from 0.31 (Checking) to 0.67 (Obsessing). Similarly, Scrupulosity was associated positively with Religious/Spiritual Struggles; the correlation coefficients ranged from 0.34 (Interpersonal) to 0.64 (Religious/Spiritual Doubt). Regarding Self-Compassion, Scrupulosity was positively correlated with lack of Self-Compassion ($r = [0.33–0.37]$; $p < 0.001$), and negatively correlated with Self-Kindness and Mindfulness ($r = [−0.21–0.23]$; $p < 0.001$). The mean correlation between Scrupulosity and OCD Symptoms was $r = 0.447$, while the mean

correlation between Scrupulosity and Religious/Spiritual Struggles was $r = 0.509$. The mean correlation between Scrupulosity and Self-Compassion was $r = -0.097$; $p$ = n.s., while the mean correlation between Scrupulosity and lack of Self-Compassion was $r = 0.348$; $p < 0.001$.

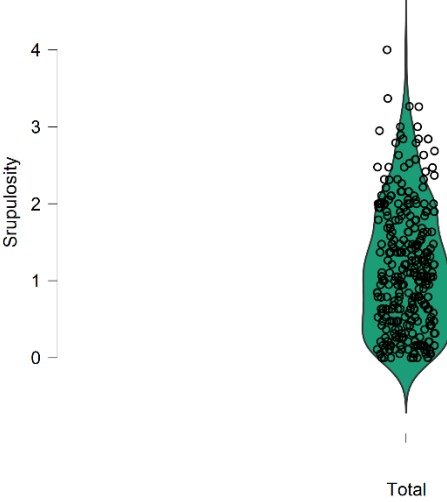

**Figure 1.** The distribution of the total score of Scrupulosity in the study (the value of cut-off level 1.42 is represented by the 64 percentile).

The mean correlation between OCD Symptoms and Religious/Spiritual Struggles was $r = 0.312$; $p < 0.001$, between OCD Symptoms and Self-Compassion was $r = -0.021$; $p$ = n.s., between OCD Symptoms and lack of Self-Compassion was $r = 0.210$; $p < 0.001$. The mean correlation between Religious/Spiritual Struggles and Self-Compassion was $r = -0.019$; $p$ = n.s., while the mean correlation between Religious/Spiritual Struggles and lack of Self-Compassion was $r = 0.260$; $p < 0.001$. The mean association between Self-Compassion and lack of Self-Compassion was $r = -0.157$; $p = 0.008$.

Comparisons between dependent correlation coefficients (Steiger 1980) indicated that the associations between Scrupulosity and OCD Symptoms did not differ from the associations between Religious/Spiritual Struggles and Scrupulosity ($Z = -1.08$; $p = 0.14$). Scrupulosity had a significantly stronger correlation with OCD Symptoms than with Self-Compassion ($Z = 6.828$; $p < 0.001$). However, Scrupulosity was correlated with OCD Symptoms similarly as with a lack of Self-Compassion ($Z = 1.519$; $p = 0.064$). Religious/Spiritual Struggles were correlated stronger with Scrupulosity than Self-Compassion ($Z = 7.764$; $p < 0.001$) and lack of Self-Compassion ($Z = 2.614$; $p = 0.004$). Scrupulosity had a stronger correlation with lack of Self-Compassion than with Self-Compassion ($Z = -5.129$; $p < 0.001$). Thus, the strongest correlations appeared between Scrupulosity, Religious/Spiritual Struggles, and OCD Symptoms, followed by a lack of Self-Compassion, and then by Self-Compassion.

**Table 2.** Correlations between the study variables.

| Variable | 1 | 2 | 3 | 4 | 5 | 6 | 7 | 8 | 9 | 10 | 11 | 12 | 13 | 14 | 15 | 16 | 17 | 18 | 19 | 20 | 21 |
|---|---|---|---|---|---|---|---|---|---|---|---|---|---|---|---|---|---|---|---|---|---|
| Self-Compassion | | | | | | | | | | | | | | | | | | | | | |
| 1. Self-Kindness | | | | | | | | | | | | | | | | | | | | | |
| 2. Self-Judgment | **−0.44** | | | | | | | | | | | | | | | | | | | | |
| 3. Common Humanity | **0.34** | **0.15** | | | | | | | | | | | | | | | | | | | |
| 4. Isolation | **−0.30** | **0.60** | **0.27** | | | | | | | | | | | | | | | | | | |
| 5. Mindfulness | **0.72** | **−0.29** | **0.39** | **−0.30** | | | | | | | | | | | | | | | | | |
| 6. Over-Identification | **−0.30** | **0.63** | **0.16** | **0.71** | **−0.36** | | | | | | | | | | | | | | | | |
| Obsessive-Compulsive Symptoms | | | | | | | | | | | | | | | | | | | | | |
| 7. Washing | **−0.07** | 0.15 | 0.15 | 0.13 | −0.03 | 0.10 | | | | | | | | | | | | | | | |
| 8. Obsessing | **−0.24** | **0.31** | 0.17 | **0.40** | **−0.22** | **0.39** | **0.46** | | | | | | | | | | | | | | |
| 9. Hoarding | −0.16 | **0.25** | 0.13 | **0.40** | **−0.19** | **0.30** | **0.39** | **0.63** | | | | | | | | | | | | | |
| 10. Ordering | −0.13 | **0.24** | 0.12 | 0.16 | −0.10 | 0.13 | **0.46** | **0.45** | **0.42** | | | | | | | | | | | | |
| 11. Checking | −0.10 | **0.22** | 0.10 | 0.14 | −0.07 | 0.16 | **0.47** | **0.44** | **0.44** | **0.45** | | | | | | | | | | | |
| 12. Neutralizing | 0.02 | 0.11 | **0.20** | 0.14 | 0.04 | 0.05 | **0.68** | **0.47** | **0.46** | **0.50** | **0.50** | | | | | | | | | | |
| Religious/Spiritual Struggles | | | | | | | | | | | | | | | | | | | | | |
| 13. Divine | −0.17 | **0.34** | 0.11 | **0.36** | −0.14 | **0.27** | **0.32** | **0.44** | **0.34** | **0.31** | **0.30** | **0.39** | | | | | | | | | |
| 14. Demonic | −0.03 | 0.18 | 0.16 | 0.18 | 0.02 | **0.19** | **0.40** | **0.41** | **0.32** | **0.34** | **0.20** | **0.45** | **0.48** | | | | | | | | |
| 15. Interpersonal | −0.11 | **0.31** | 0.15 | **0.24** | −0.07 | 0.17 | **0.30** | **0.33** | **0.27** | **0.27** | **0.27** | **0.35** | **0.57** | **0.35** | | | | | | | |
| 16. Moral | −0.09 | **0.26** | 0.16 | **0.26** | −0.11 | **0.25** | 0.13 | **0.37** | **0.32** | **0.20** | 0.16 | 0.18 | **0.39** | **0.38** | **0.41** | | | | | | |
| 17. Doubt | **−0.19** | **0.31** | 0.17 | **0.30** | −0.15 | **0.28** | **0.29** | **0.50** | **0.41** | **0.26** | **0.27** | **0.27** | **0.56** | **0.41** | **0.50** | **0.51** | | | | | |
| Religiosity | | | | | | | | | | | | | | | | | | | | | |
| 18. Religious identification | 0.08 | 0.01 | −0.07 | −0.05 | 0.03 | 0.12 | **−0.25** | −0.13 | −0.03 | −0.17 | −0.14 | **−0.22** | −0.16 | 0.06 | **−0.25** | 0.16 | −0.07 | | | | |
| 19. Religious identity | 0.07 | 0.05 | −0.04 | 0.03 | 0.05 | 0.12 | **−0.23** | −0.15 | −0.03 | −0.16 | −0.16 | **−0.19** | −0.08 | 0.06 | **−0.21** | 0.16 | −0.06 | **0.86** | | | |
| 20. Religious attendance | −0.02 | 0.04 | −0.08 | −0.01 | −0.05 | 0.09 | **−0.20** | −0.04 | 0.02 | −0.10 | −0.12 | **−0.22** | −0.08 | 0.03 | **−0.20** | **0.23** | −0.01 | **0.71** | **0.70** | | |
| Scrupulosity | | | | | | | | | | | | | | | | | | | | | |
| 21. Fear of Sin | **−0.24** | **0.33** | 0.17 | **0.37** | **−0.23** | **0.35** | **0.42** | **0.67** | **0.54** | **0.40** | **0.31** | **0.46** | **0.47** | **0.53** | **0.35** | **0.56** | **0.62** | −0.04 | −0.03 | 0.03 | |
| 22. Fear of God | **−0.21** | **0.34** | 0.16 | **0.36** | **−0.23** | **0.34** | **0.37** | **0.60** | **0.50** | **0.37** | **0.31** | **0.41** | **0.52** | **0.50** | **0.34** | **0.56** | **0.64** | 0.01 | 0.05 | 0.09 | **0.87** |

Note. Correlation coefficients higher than |0.19| are significant at $p < 0.001$ (in bold).

### 3.2. A Network Analysis

In the next step, we performed a network analysis. The number of non-zero edges was 110, and the sparsity of the network was 0.524. The regularized partial correlation network based on the complete cases is presented in Figure 2.

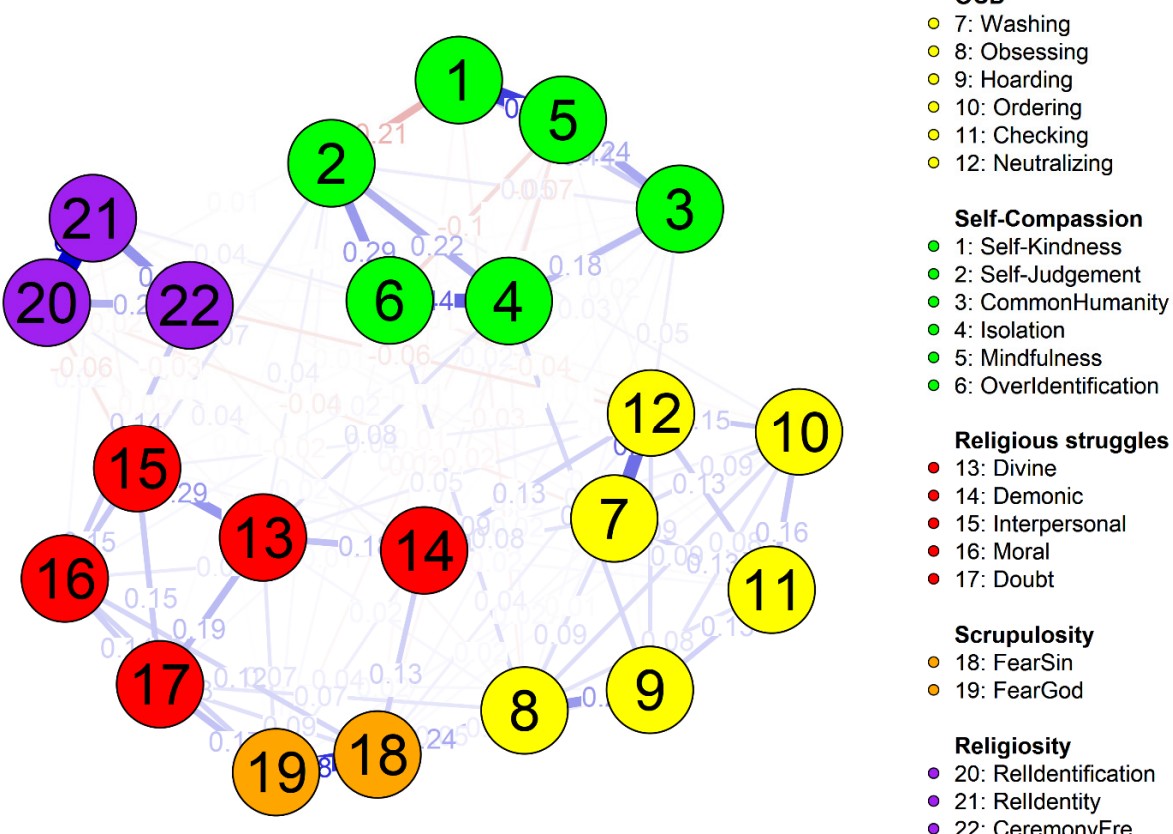

**Figure 2.** The regularized partial correlation network. (Different colors represent different groups of symptoms according to the legend; blue edges represent positive, while red edges represent negative associations between nodes; numbers indicate partial correlations between nodes.).

Based on the 95% bootstrapped CI, the edge-weights and node strengths appeared stable (Appendix A; Figures A1–A3). Centrality measures are given in Table 3 and in Figure 3. Expected influence of symptoms of scrupulosity was also the highest in the network.

**Table 3.** Centrality measures of the network analysis.

| Variable | Betweenness | Closeness | Strengths | Bridge Strength | BEI |
|---|---|---|---|---|---|
| Self-Compassion | | | | | |
| Self-Kindness | −0.455 | −1.260 | 0.016 | - | - |
| Self-Judgment | 0.816 | −0.126 | −0.018 | - | - |
| Common Humanity | −0.535 | −0.945 | −1.026 | - | - |
| Isolation | 1.531 | 0.468 | 0.962 | - | - |
| Mindfulness | −1.091 | −1.477 | 0.543 | - | - |
| Over-Identification | 0.498 | 0.399 | 0.380 | - | - |
| Obsessive-Compulsive Symptoms | | | | | |
| Washing | −1.250 | −0.327 | −0.406 | 0.097 | 0.290 |
| Obsessing | 1.849 | 1.714 | 0.679 | 0.334 | 0.683 |
| Hoarding | 0.498 | 1.132 | −0.669 | 0.120 | 0.359 |
| Ordering | −1.329 | −0.780 | −1.426 | 0.068 | 0.206 |
| Checking | −1.329 | −0.454 | −1.507 | 0.027 | 0.144 |
| Neutralizing | 0.578 | 0.283 | 1.467 | 0.251 | 0.473 |

**Table 3.** *Cont.*

| Variable | Betweenness | Closeness | Strengths | Bridge Strength | BEI |
|---|---|---|---|---|---|
| Religious/Spiritual Struggles | | | | | |
| Divine | 0.896 | 0.905 | −0.002 | 0.156 | 0.427 |
| Demonic | −0.376 | 0.328 | −1.135 | 0.421 | 0.809 |
| Interpersonal | 0.260 | 0.638 | −0.529 | 0.045 | 0.228 |
| Moral | 0.737 | 0.223 | −0.872 | 0.243 | 0.517 |
| Doubt | −0.693 | 0.496 | −0.518 | 0.353 | 0.692 |
| Religiosity | | | | | |
| Religious identification | −1.329 | −1.596 | 0.939 | - | - |
| Religious identity | −0.773 | −1.396 | 1.134 | - | - |
| Religious attendance | 0.498 | −0.927 | −1.030 | - | - |
| Scrupulosity | | | | | |
| Fear of Sin | 1.452 | 1.556 | 2.091 | 0.692 | 1.381 |
| Fear of God | −0.455 | 1.147 | 0.925 | 0.478 | 1.146 |

Note, BEI—bridge expected influence.

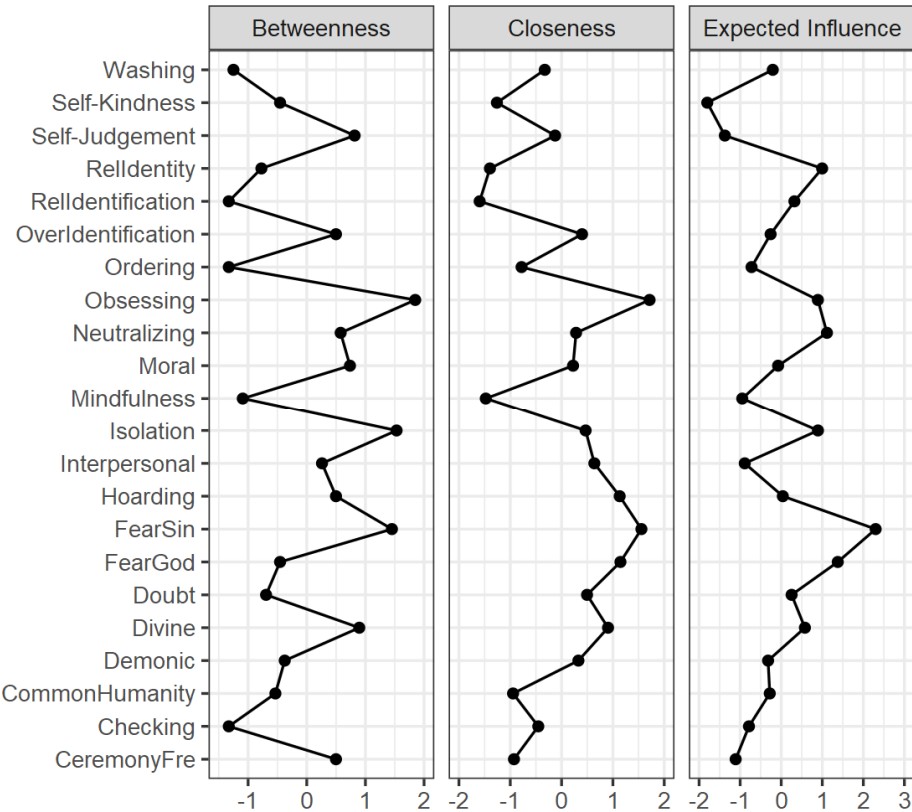

**Figure 3.** Centrality plot for the study network.

According to the centrality measures, the most central symptoms in the network could be regarded as Fear of Sin (strength = 2.091; betweenness = 1.452), Obsessing (strength = 0.679; betweenness = 1.849) and Neutralizing (strength = 1.467; betweenness = 0.587), and Isolation (strength = 0.962; betweenness = 1.531). Regarding closeness, the symptoms which can reach any other symptom in the network the fastest are Obsessing (closeness = 1.714) and Fear of Sin (closeness = 1.556; see also Figure A4 in Appendix A).

The estimation of the CS-coefficient indicated that node strengths were stable under subsetting cases (*C S* [cor = 0.7] = 0.596) and reach the cutoff of 0.5 from our simulation study required to consider the metric stable (Epskamp et al. 2018). The estimation of the CS-coefficient indicated that edges were stable under subsetting cases (*C S* [cor = 0.7] = 0.750) and reach the cutoff of 0.5 from our simulation study required to consider the metric stable (Epskamp et al. 2018).

The strongest edge-weights appeared between symptoms of Scrupulosity (0.584), between Fear of Sin and Obsessing (0.242), Religious/Spiritual Struggles and Fear of Sin

(0.119), Religious/Spiritual Struggles and Fear of God (0.126), and between Fear of God and (religious) Doubt (0.174) and between Demonic Struggles and Fear of Sin (0.132). All edge-weights are given in Table A1 in the Appendix A.

Finally, we estimated bridge symptoms between the symptoms of Religious/Spiritual Struggles, Scrupulosity and OCD Symptoms, due to the fact that the strongest connections appeared in correlation and network analysis (Table 3). We calculated two indices of bridge symptoms: (a) bridge strength (the sum of the absolute value of all edges that exist between a given node and all nodes that are not in the same community as a given node), and (b) bridge expected influence two-step, which considers both the sum of the value [+ or −] of all edges that exist between a given node and all nodes that are not in the same community as a given node and the indirect effect that a given node may have on other communities through other nodes (Kaiser et al. 2021). For example, the bridge expected influence of Religious/Spiritual Struggle will reflect how a given struggle may be directly related to Scrupulosity and OCD Symptoms, and also indirectly related to OCD symptoms via the Scrupulosity symptoms (and vice versa). The bridge strength and bridge expected influence were stable in the present analysis (*C S* [cor = 0.70] = 0.671, and *C S* [cor = 0.70] = 0.671, respectively).

The analysis indicated that both dimensions of Scrupulosity had the highest bridge strength and bridge expected influence. Among Religious/Spiritual Struggles, the Demonic struggles had the highest bridge strength and bridge expected influence, but Moral struggles and (religious) Doubt had relatively high bridge expected influence. Thus, these three religious struggles may indirectly affect other symptoms included in the analysis, namely Scrupulosity and OCD Symptoms. Among OCD symptoms, Obsessing appeared to have the highest bridge strength and expected influence. According to all results, Moral struggles, (religious) Doubt and Demonic struggles could be regarded as bridge symptoms between Religious/Spiritual Struggles, Scrupulosity and OCD Symptoms. Scrupulosity could also be generally regarded as containing bridge symptoms between Religious/Spiritual Struggle and OCD Symptoms (mostly Obsessing). Obsessing could be regarded as a bridge symptom between OCD Symptoms, Scrupulosity and Religious/Spiritual Struggles.

The spin glass algorithm indicated six groups of symptoms. The first one included symptoms of lack of Self-Compassion (i.e., Self-Judgement, Isolation, and Overidentification), the second one included Obsessing and Hoarding, the third one included other OCD symptoms (Washing, Checking, and Neutralizing). The fourth community consisted of Self-Compassion (i.e., Self-Kindness, Common Humanity, and Mindfulness), and the fifth community consisted of Religiosity items (i.e., identification with religious beliefs, the role of religion in identity, and religious attendance). The last community included all types of religious and spiritual struggles, but also two factors of Scrupulosity. These results indicated, therefore, that scruples could be a part of normative religious struggles, rather than an independent OCD-related religious phenomenon. To test this hypothesis, we used SEM to investigate whether scrupulosity and religious/spiritual struggles were loaded by one latent variable. The one-factor model had poor fit to data ($\chi^2$ = 152.436; *df* = 14; *p* < 0.001; CFA = 0.871; TLI = 0.810; RMSEA = 0.184; SRMR = 0.091), and was worse than the fit of the two-factor model ($\Delta \chi^2$ = 76.132; *df* = 1; *p* < 0.001). Thus, religious struggles and scrupulosity were associated and could be activated mutually, but these variables were not underlain by a common factor.

### 3.3. Structural Equation Modeling

In the last step, we performed a SEM analysis of Self-Compassion, lack of Self-Compassion, Religiosity, Religious/Spiritual Struggles and OCD Symptoms in predicting scrupulosity. Mardia's coefficients (skeweness = 65.091; *p* < 0.001; kurtosis = 582.786; *p* < 0.001) indicated multivariate non-normality of data. Thus, we used the diagonally weighted least square (DWLS) estimator (Li 2016). The model fit the data well ($\chi^2$ = 451.108; *df* = 194; *p* < 0.001; CFA = 0.951; TLI = 0.942; RMSEA = 0.067; SRMR = 0.085). All observed variables had a high and significant loading on the respective latent variables ($\lambda$ = [0.597; 0.974]). The structural model is given in Figure 4.

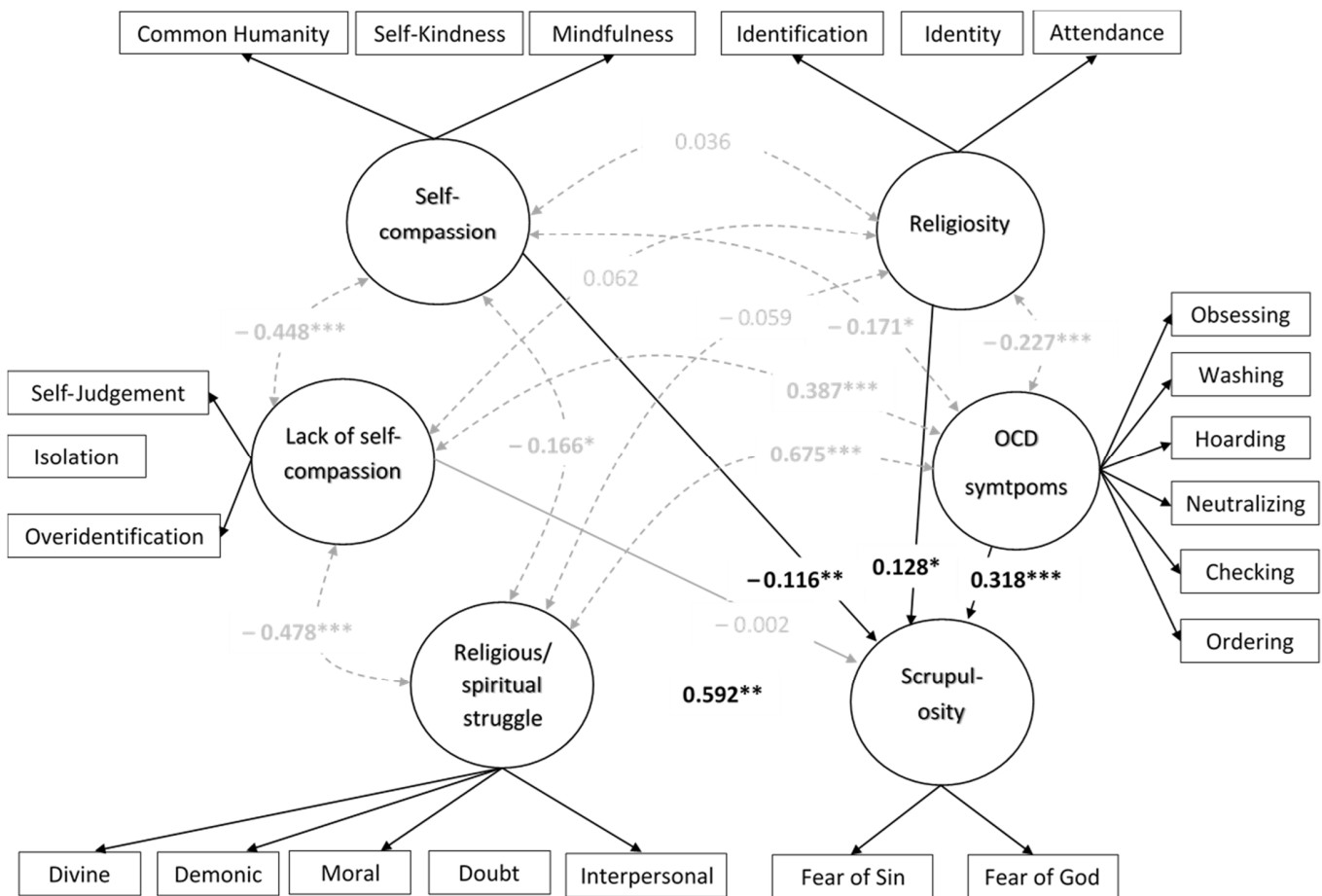

**Figure 4.** The structural model for the study variables (gray and dotted paths indicate covariations; bold coefficients were statistically significant; the coefficients are standardized; for the sake of clarity the loadings of observed variables on latent variables were omitted). * $p < 0.05$; ** $p < 0.01$; *** $p < 0.001$.

The structural model explained 74.1% of the variance in Scrupulosity. Religious/ Spiritual Struggles was the strongest predictor of Scrupulosity ($\beta$ = 0.592; $p < 0.001$), followed by OCD Symptoms ($\beta$ = 0.318; $p < 0.001$). Religiosity was positively, but weakly associated with Scrupulosity ($\beta$ = 0.128; $p = 0.003$), while Self-Compassion was associated with lower Scrupulosity ($\beta$ = −.116; $p = 0.010$). A significant positive covariance appeared between OCD Symptoms and Religious/Spiritual Struggles ($\beta$ = 0.675; $p < 0.001$), lack of Self-Compassion and Religious/Spiritual Struggles ($\beta$ = 0.478; $p < 0.001$), and between lack of Self-Compassion and OCD Symptoms ($\beta$ = 0.387; $p < 0.001$). Religious identification, attendance and the high role of religion in one's identity were negatively correlated with OCD Symptoms ($\beta$ = −0.227; $p < 0.001$), but their associations with Religious/Spiritual Struggles were non-significant.

## 4. Discussion

The goal of the study was to investigate scrupulosity in the network of religious struggles, OCD symptoms, religiosity, and self-compassion. Inspection of correlation analysis and network analysis indicated that scrupulosity is correlated with lower self-compassion (self-judgement, isolation, and over-identification), but primarily with religious/spiritual struggles, and OCD symptoms. The last result was in line with previous findings demonstrating close relationships between obsessive-compulsive symptoms and scrupulosity (Abramowitz and Buchholtz 2020). Positive associations between religious/spiritual struggles and scrupulosity indicated that internal conflicts due to the religion could be regarded as reflections or sources of scrupulosity. This finding is in line with recent studies indicating that religious crisis, but not fundamentalism, was correlated with scrupulosity (Henderson

et al. 2022). The community examination using the spin glass algorithm in the network analysis demonstrated that Scrupulosity was a part of the same symptom community as Religious/Spiritual Struggles. However, SEM indicated that this association was not a result of a common latent process. Thus, religious struggles and scrupulosity seem to activate each other in the network, but are not the reflections of the same psychological religious process. Religious scruples could be correlated with religious doubts and struggles, but scrupulosity is not the next type of religious struggles. This finding is important from the clinical perspective. Most scrupulous individuals perceive their symptoms as interfering with their religious experience (Siev et al. 2011). Thus, clinicians should help their clients to differentiate between normative religious doubts and scruples. It is also important to avoid normalization of religious scruples by religious communities (Abramowitz and Buchholtz 2020). Future studies should also investigate the direction of associations between scrupulosity and religious struggles using longitudinal designs.

Scrupulosity was also found as a correlation of low self-compassion, which is consistent with previous studies (Borgogna et al. 2020; Fisak et al. 2019). This result indicates that a lack of self-kindness and poor insight could foster scrupulosity among religious people (Tolin et al. 2001). Contrarily, isolation, self-judgement and over-identification could also create a ground for a rigid, lacking self-forgiveness approach to one's sin (Brodar et al. 2015). SEM demonstrated that scrupulosity was frequent among individuals with lower self-compassion, but the association was weak. Similarly, network analyses indicated a relatively weak association between self-compassion and scrupulosity. These results suggest that treatment of scrupulosity could benefit from introducing self-compassion exercises, but focusing only on the development of the self-compassionate attitude seems to be not enough to treat scruples.

Contrary to expectations, religiosity did not correlate with scrupulosity (see Abramowitz and Buchholtz 2020). Although a number of studies demonstrated positive associations between religiosity and religious scruples (Siev et al. 2021), some previous studies did not show significant associations between these variables (Nelson et al. 2006). The associations between religiosity and scrupulosity, therefore, seem to be complex and could depend on the particular cultural context (Abramowitz and Jcoby 2014). Also, contrary to the previous findings concerning associations between religiosity and obsessiveness (Inozu et al. 2012), religiosity correlated negatively with OCD symptoms. However, previous findings demonstrated that religious fundamentalism positively correlated with compulsiveness and obsessiveness (Inozu et al. 2012). In the present study, we did not measure such a dimension of religiosity as fundamentalism. Thus, it is possible that only a particularly rigid and extreme religiosity fosters OCD symptomatology, while other dimensions of religiosity could be differently related to OCD symptoms. Although the correlational and network analysis in the present study did not demonstrate significant associations between Religiosity and Scrupulosity, SEM showed a positive yet weak association. This result could suggest that the latent variable reflecting Religiosity is associated with a higher probability of religious scruples. However, this finding could also be a result of a negative covariation of Religiosity and OCD Symptoms, which were positively associated with scrupulosity. Religiosity could play some role in scruples, but scruples were not a simple consequence of religiosity itself (Abramowitz and Buchholtz 2020). Future studies should use a more precise measure of religiosity to examine these associations in more detail. From the practical point of view, our findings indicate that religious scruples could appear among people who did not perceive themselves as particularly religious in terms of religious attendance and the role of religiosity in their identity. A fear of hell or condemnation could be a serious reason of psychological distress and compulsions among non-religious individuals as well.

The present study showed that scrupulosity is positioned on the intersection between the mental experiences associated with tension, strain, and conflicts about sacred matters (Exline et al. 2014) and an obsessing reflecting OCD symptomatology. The bridge analysis indicated that demonic, moral, and religious doubt struggles, and obsessing, as well as scrupulosity were all positively correlated. In the network analysis approach, this may

indicate that there could be an internal dynamic between these symptoms, and one symptom could activate others. Similarly, SEM demonstrated that Religious/Spiritual Struggles and OCD Symptoms were positively associated, and both predicted higher Scrupulosity. From the clinical point of view, it could indicate that strong religious struggles may activate scrupulosity which could turn into obsessing. This could illustrate a process of developing pathologized religiosity out of religious struggles experienced by individuals without proper insight and a healthy attitude toward the self. The reverse pattern is also possible, namely when OCD-related obsessions "invade" a sphere of religiosity. Although such causal interpretations of the current results are unjustified, future studies should investigate the possibility of causal relationships between these symptoms in longitudinal approach. The present study, however, showed that religious scruples seem to be more closely linked to religious struggles than OCD symptoms. This finding could encourage clinicians to pay more attention in conversation with their clients to foster their insight into boundaries of normative religious struggles and to develop their knowledge about characteristics of religious scruples. These findings also indicated that patients with an inclination to develop obsessive symptomatology (e.g., due to personality factors such as difficulty with change or maladaptive perfectionism; Fang et al. 2016; Siev et al. 2021) could be at a higher risk of developing religious scruples due also to their stronger religious struggles.

Both the network analysis approach and the SEM approach yielded similar results concerning scrupulosity. When focusing on the associations between symptoms or latent variables, scrupulosity appeared as a phenomenon at the intersection between normative religious struggle and OCD symptomatology. The network analysis demonstrated that the central among these symptoms are Fear of Sin and Obsessing. Thus, these symptoms should be the targets of treatment of individuals suffering from religious scruples. This result also demonstrated the advantage of the network analysis which allows us to detect the particular symptoms which could strongly activate other symptoms in the studied network. A combination of both statistical methods indicated that although religious struggles and scrupulosity belong to one community of symptoms, the processes underlying these symptoms are different. Thus, scruples are probably not simply religious doubts and should not be disregarded in the examination of clients' religiosity. From the methodological point of view, future studies on the psychopathology of scruples could benefit from the combined methodology of the network analysis and the latent structures analysis. Future studies should examine more precisely the role of religiosity in the studies network. SEM indicated that religiosity could be associated with higher scrupulosity, while the network analysis indicated that religiosity and scruples were relatively independent.

The present study had some limitations. First, the cross-sectional design forbids any causal interpretations of our data. Second, the participants were not representative of the population of religious individuals in Poland. Although recruitment through social media is suitable for populations that are difficult to reach and could help participants in maintaining anonymity, this sampling method is not without limitations (Topolovec-Vranic and Natarajan 2016). Thus, future studies should investigate the associations between studied variables in representative samples and in direct contact. Third, the measurement of religiosity was limited. We used only three items to measure identification with religion, the role of religion for one's identity, and the attendance of religious ceremonies. These items were very similar to the measurement of numerous studies on the association between religiosity and scrupulosity (Siev et al. 2011) and between religiosity and other psychological processes, e.g., forgiveness (Fincham and May 2019). However, a more in-depth analysis of various dimensions of religiosity (e.g., centrality of religiosity) could allow for a better investigation into the associations between the role of religion for an individual and that individual's scruples.

## 5. Conclusions

The present study allows us to describe a psychological profile of the scrupulous faithful as individuals strongly experiencing moral, religious, and demonic struggles.

A self-reference of scrupulous individuals is lacking self-kindness and insight, and this may lead to isolation, over-identification and a self-judgmental attitude. The association between fear of sin and fear of God, which indicate two basic dimensions of scrupulosity, and obsessing were highly positive. This indicates that scrupulosity may pass relatively smoothly to obsession.

These findings seem very important for psychological and pastoral counselling for religious individuals. Individuals with scrupulosity need both pastoral assistance in coping with religious struggles, but also clinical counselling to prevent scruples from developing into obsessions. Moreover, general support of their insight and a more self-compassionate attitude toward themselves seem necessary.

**Author Contributions:** Conceptualization, M.M., M.B.-M. and K.M.; resources, M.M., M.B.-M., K.M.; methodology, M.M.; formal analysis, M.M.; investigation, M.M., M.B.-M. and K.M.; data curation, M.M.; writing—original draft preparation, M.M.; writing—review and editing, M.M., M.B.-M. and K.M.; visualization, M.M. All authors have read and agreed to the published version of the manuscript.

**Funding:** This research received no external funding.

**Institutional Review Board Statement:** The study was conducted according to the guidelines of the Declaration of Helsinki, and approved by the Ethics Committee of the University of Silesia in Katowice, Poland (KEUS 244/04.2022).

**Informed Consent Statement:** Informed consent was obtained from all subjects involved in the study.

**Data Availability Statement:** The data are available at: https://osf.io/d5h7n/?view_only=f1b18166 6bd347c59a170887c1dec340 (accessed on 17 September 2022).

**Acknowledgments:** The authors would like to thank the participants and research assistants for their contribution to this study.

**Conflicts of Interest:** The authors declare no conflict of interest.

**Appendix A**

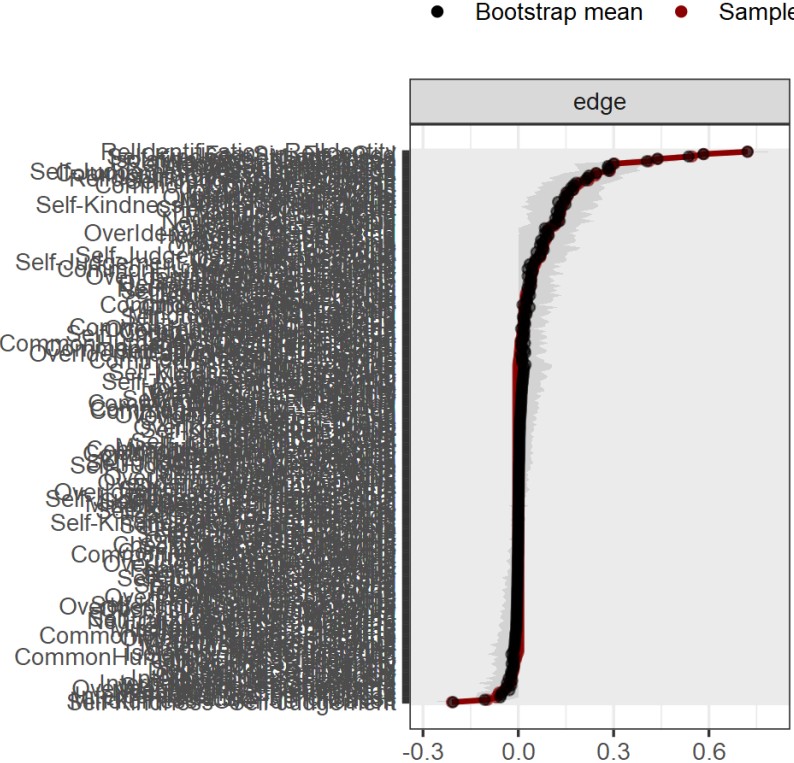

**Figure A1.** 95% confidence intervals for edges.

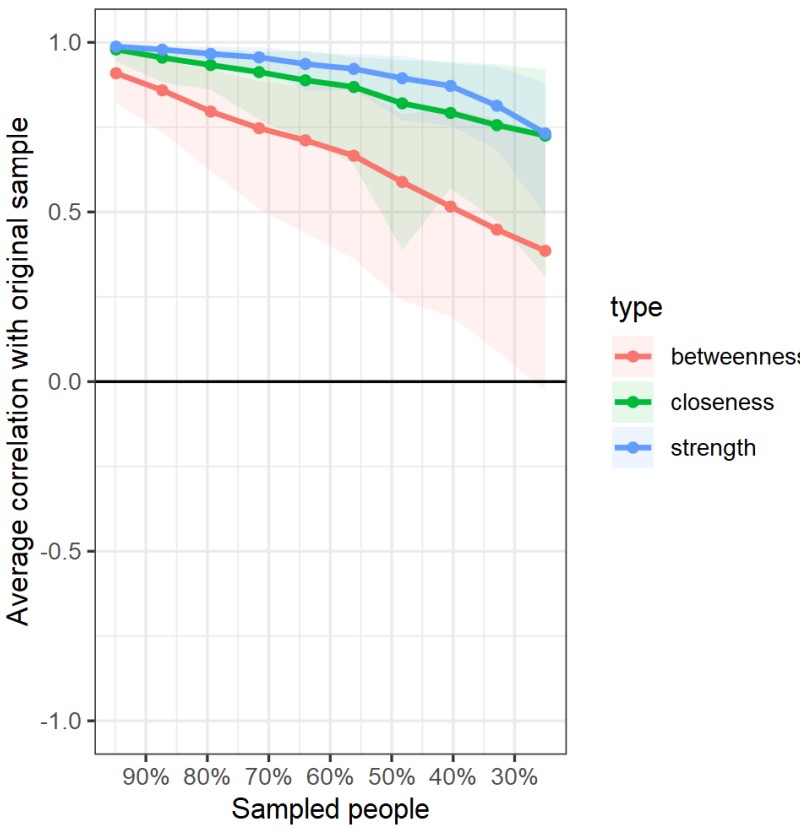

**Figure A2.** Centrality stability.

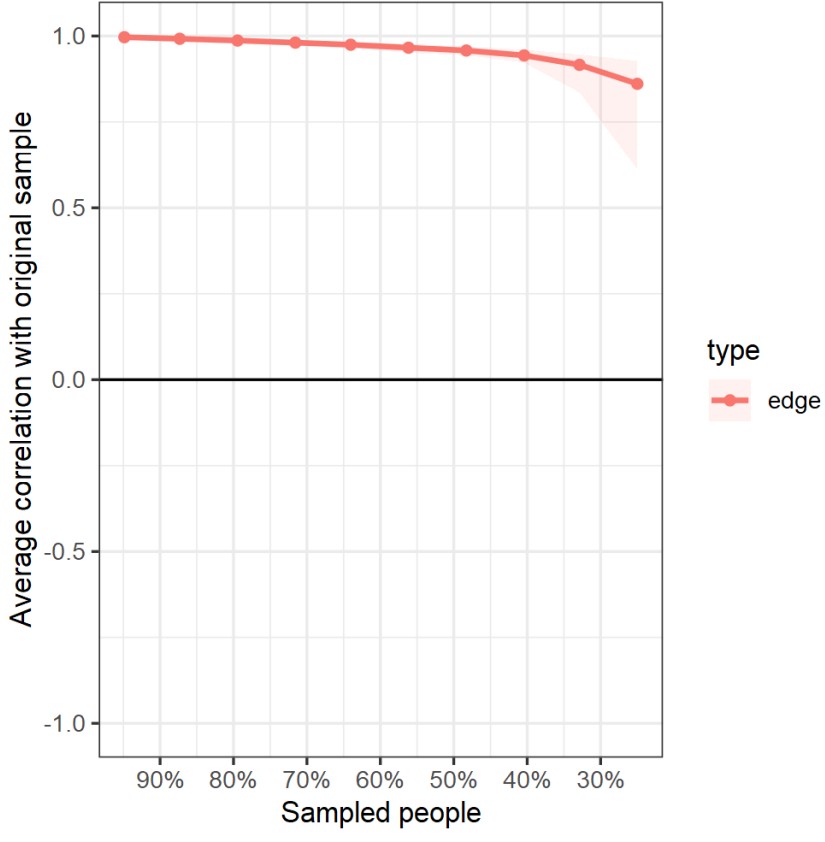

**Figure A3.** Edge stability.

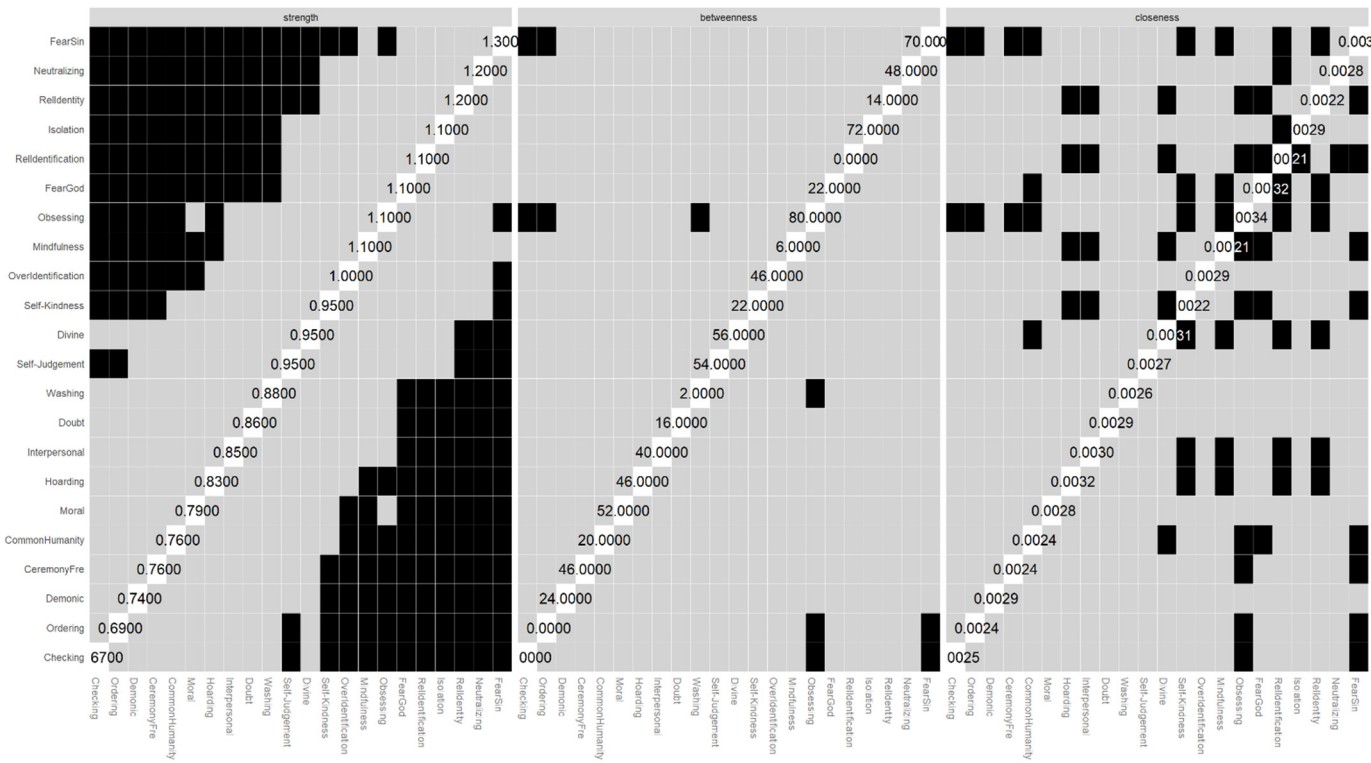

**Figure A4.** Differences in centrality of nodes.



**Table A1.** Edge-weights of the network analysis.

| Variable | Self-Kindness | Self-Judgement | Common Humanity | Isolation | Mindfulness | Over Identi-fication | Washing | Obsessing | Hoarding | Ordering | Checking | Neutralizing | Divine | Demonic | Interpersonal | Moral | Doubt | FearSin | FearGod | RelIdenti fication | RelIdentity | Ceremony Fre |
|---|---|---|---|---|---|---|---|---|---|---|---|---|---|---|---|---|---|---|---|---|---|---|
| Self-Kindness | 0.000 | −0.209 | 0.141 | −0.003 | 0.547 | 0.000 | 0.000 | −0.028 | 0.000 | 0.000 | 0.000 | 0.000 | 0.000 | 0.000 | 0.000 | 0.000 | 0.000 | −0.015 | 0.000 | 0.013 | 0.000 | 0.000 |
| Self-Judgement | −0.209 | 0.000 | 0.054 | 0.221 | 0.000 | 0.288 | 0.000 | 0.000 | 0.000 | 0.030 | 0.005 | 0.000 | 0.040 | 0.000 | 0.070 | 0.007 | 0.007 | 0.000 | 0.018 | 0.000 | 0.000 | 0.000 |
| CommonHumanity | 0.141 | 0.054 | 0.000 | 0.184 | 0.244 | 0.007 | 0.000 | 0.015 | 0.000 | 0.000 | 0.000 | 0.047 | 0.000 | 0.002 | 0.006 | 0.023 | 0.014 | 0.020 | 0.000 | −0.006 | 0.000 | 0.000 |
| Isolation | −0.003 | 0.221 | 0.184 | 0.000 | −0.070 | 0.439 | 0.000 | 0.018 | 0.113 | 0.000 | 0.000 | 0.000 | 0.078 | 0.000 | 0.000 | 0.000 | 0.000 | 0.004 | 0.000 | 0.000 | 0.000 | 0.000 |
| Mindfulness | 0.547 | 0.000 | 0.244 | −0.070 | 0.000 | −0.097 | 0.000 | 0.000 | −0.007 | 0.000 | 0.000 | 0.018 | 0.000 | 0.025 | 0.000 | 0.000 | 0.000 | −0.023 | −0.021 | 0.000 | 0.000 | 0.000 |
| OverIdentification | 0.000 | 0.288 | 0.007 | 0.439 | −0.097 | 0.000 | 0.000 | 0.094 | 0.000 | 0.000 | 0.000 | −0.037 | 0.000 | 0.000 | 0.000 | 0.000 | 0.000 | 0.000 | 0.015 | 0.005 | 0.040 | 0.000 |
| Washing | 0.000 | 0.000 | 0.000 | 0.000 | 0.000 | 0.000 | 0.000 | 0.086 | 0.000 | 0.095 | 0.132 | 0.412 | 0.000 | 0.081 | 0.000 | 0.000 | 0.000 | 0.016 | 0.000 | −0.044 | −0.012 | 0.000 |
| Obsessing | −0.028 | 0.000 | 0.015 | 0.018 | 0.000 | 0.094 | 0.086 | 0.000 | 0.288 | 0.095 | 0.078 | 0.019 | 0.023 | 0.000 | 0.000 | 0.000 | 0.072 | 0.242 | 0.000 | 0.000 | −0.019 | 0.000 |
| Hoarding | 0.000 | 0.000 | 0.000 | 0.113 | −0.007 | 0.000 | 0.000 | 0.288 | 0.000 | 0.077 | 0.127 | 0.094 | 0.000 | 0.000 | 0.000 | 0.000 | 0.023 | 0.045 | 0.054 | 0.000 | 0.000 | 0.000 |
| Ordering | 0.000 | 0.030 | 0.000 | 0.000 | 0.000 | 0.000 | 0.095 | 0.095 | 0.077 | 0.000 | 0.157 | 0.153 | 0.001 | 0.037 | 0.000 | 0.000 | 0.000 | 0.012 | 0.018 | 0.000 | −0.013 | 0.000 |
| Checking | 0.000 | 0.005 | 0.000 | 0.000 | 0.000 | 0.000 | 0.132 | 0.078 | 0.127 | 0.157 | 0.000 | 0.134 | 0.017 | 0.000 | 0.010 | 0.000 | 0.000 | 0.000 | 0.000 | 0.000 | −0.012 | 0.000 |
| Neutralizing | 0.000 | 0.000 | 0.047 | 0.000 | 0.018 | −0.037 | 0.412 | 0.019 | 0.094 | 0.153 | 0.134 | 0.000 | 0.048 | 0.126 | 0.036 | 0.000 | 0.000 | 0.040 | 0.000 | 0.000 | 0.000 | −0.061 |
| Divine | 0.000 | 0.040 | 0.000 | 0.078 | 0.000 | 0.000 | 0.000 | 0.023 | 0.000 | 0.001 | 0.017 | 0.048 | 0.000 | 0.175 | 0.291 | 0.000 | 0.193 | 0.000 | 0.070 | −0.016 | 0.000 | 0.000 |
| Demonic | 0.000 | 0.000 | 0.002 | 0.000 | 0.025 | 0.000 | 0.081 | 0.000 | 0.000 | 0.037 | 0.000 | 0.126 | 0.175 | 0.000 | 0.023 | 0.060 | 0.000 | 0.132 | 0.042 | 0.038 | 0.000 | 0.000 |
| Interpersonal | −0.028 | 0.070 | 0.006 | 0.000 | 0.000 | 0.000 | 0.000 | 0.000 | 0.000 | 0.000 | 0.010 | 0.036 | 0.291 | 0.023 | 0.000 | 0.154 | 0.149 | 0.000 | 0.000 | −0.063 | −0.022 | −0.028 |
| Moral | 0.000 | 0.007 | 0.023 | 0.000 | 0.000 | 0.000 | 0.000 | 0.000 | 0.000 | 0.000 | 0.000 | 0.000 | 0.000 | 0.060 | 0.154 | 0.000 | 0.137 | 0.119 | 0.126 | 0.000 | 0.019 | 0.144 |
| Doubt | 0.000 | 0.007 | 0.014 | 0.000 | 0.000 | 0.000 | 0.000 | 0.072 | 0.023 | 0.000 | 0.000 | 0.000 | 0.193 | 0.000 | 0.149 | 0.137 | 0.000 | 0.088 | 0.173 | 0.000 | 0.000 | 0.000 |
| FearSin | −0.015 | 0.000 | 0.020 | 0.004 | −0.023 | 0.000 | 0.016 | 0.242 | 0.045 | 0.012 | 0.000 | 0.040 | 0.000 | 0.132 | 0.000 | 0.119 | 0.088 | 0.000 | 0.584 | 0.000 | 0.000 | 0.000 |
| FearGod | 0.000 | 0.018 | 0.000 | 0.000 | −0.021 | 0.015 | 0.000 | 0.000 | 0.054 | 0.018 | 0.000 | 0.000 | 0.070 | 0.042 | 0.000 | 0.126 | 0.173 | 0.584 | 0.000 | 0.000 | 0.000 | 0.003 |
| RelIdentification | 0.013 | 0.000 | −0.006 | 0.000 | 0.000 | 0.005 | −0.044 | 0.000 | 0.000 | 0.000 | 0.000 | 0.000 | −0.016 | 0.038 | −0.063 | 0.000 | 0.000 | 0.000 | 0.000 | 0.000 | 0.721 | 0.221 |
| RelIdentity | 0.000 | 0.000 | 0.000 | 0.000 | 0.000 | 0.040 | −0.012 | −0.019 | 0.000 | −0.013 | −0.012 | 0.000 | 0.000 | 0.000 | −0.022 | 0.019 | 0.000 | 0.000 | 0.000 | 0.721 | 0.000 | 0.303 |
| CeremonyFre | 0.000 | 0.000 | 0.000 | 0.000 | 0.000 | 0.000 | 0.000 | 0.000 | 0.000 | 0.000 | 0.000 | −0.061 | 0.000 | 0.000 | −0.028 | 0.144 | 0.000 | 0.000 | 0.003 | 0.221 | 0.303 | 0.000 |

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
