# Peer review of "Scrupulosity in the Network of Obsessive-Compulsive Symptoms, Religious Struggles, and Self-Compassion: A Study in a Non-Clinical Sample"

_religions, doi:10.3390/rel13100879_

Round 1

Author Response

Response to the Reviewer#1

Dear Reviewer,

Thank you for all insightful suggestions regarding the previous version of the manuscript. Below, you will find our responses to all your suggestions.

Thank you very much for your effort put in the correction of the previous version of our manuscript. We hope that changes that we made address correctly your suggestions.

The Authors

Reviewer #1: I would like to express my grateful feeling for the opportunity to share my thoughts about this article submitted to Religions. I have read the article carefully to understand the main idea about the position of scrupulosity within the network obsessive compulsive symptoms and human religiosity. The comes with the fresh idea about the bridge of human religiosity and psychology in which scrupulosity is examined within this framework. Across the manuscript, I found that the concepts involved in the study were well-defined and clearly explained, however, I also think that the argumentation that links one concept and the others was lack of clarity. For example, when reading the introduction of the manuscript, I think there is a lack of theoretical explanation on the association of self-compassion and scrupulosity. Therefore, for this part, I would suggest the author(s) to dive more deeply in the previous studies that might link self-compassion with scrupulosity or any closely related concepts. 

The Authors: Thank you for this suggestion. In the revised form we discussed the associations between scrupulosity and self-compassion in more detail. We focused on the associations between self-compassion and mindfulness with processes common in scrupulosity (namely, thought-action fusion and intolerance of uncertainty). We showed that self-compassion is associated with lower thought-action fusion and intolerance of uncertainty. Moreover, we showed that scrupulosity is associated with religious perfectionism. Self-compassion was also associated with less perfectionism. Finally, we refer to studies that directly showed that nonjudging, mindfulness, and self-compassion were associated with lower scrupulosity. The modified sentences was included below:

lines 178-201 (in the manuscript with changes tracked): Previous studies found that scrupulosity was correlated positively with thought–action fusion (Siev et al., 2017) and intolerance of uncertainty (Nelson et al., 2006). Self-compassion and mindfulness were associated with a lower intolerance of uncertainty (Mantzios et al. 2015). Mindfulness-based interventions were effective in reduction of thought-action fusion and intolerance of uncertainty among OCD patients (Asli Azad et al. 2019). Thus, self-compassionate and mindfulness individuals could avoid the development of religious scruples due to their higher tolerance of uncertainty and lower thought-action fusion. Self-compassion was also correlated with lesser maladaptive perfectionism (Stoeber, Lalova and Lumley 2020). Among religious individuals, self-compassion was associated with less perfectionism and with higher perceived forgiveness by God and received support (Brodar, Crosskey and Thompson 2015). Scrupulosity represents a strong fear of immorality (Huppert and Fradkin, 2016) that could be regarded as a reflection of religious perfectionism which is lower among self-compassionate individuals (Brodar et al., 2015). Research studies on the direct association between mindfulness and scrupulosity are rare. However, they indicate that mindful individuals experience fewer religious scruples. Mindfulness, but particularly nonjudging, was negatively correlated with scrupulosity in a sample of undergraduate students (Fisak et al., 2019). Self-compassion was negatively associated with scrupulosity among men (Borgogna et al., 2020). These results suggest that scrupulosity develops easily among less mindful and more self-judging individuals.

Reviewer #1: The concept of scrupulosity which was operationally defined through two dimensions (i.e., fear of sin and fear of God) did not represent the extreme feeling that might be contained by the concept. In my mind, the fear of God, as well as the fear of sin are the logical consequence of the religiosity itself. Without being too scrupulous, a religious person must have those feelings as the foundation of their morality which is sourced from their faith. Therefore, I would suggest to re-frame the description of scrupulosity to represent more deviant pattern of feeling (which implied on behaviours) within human religiosity.

The Authors: The operationalization of scrupulosity was based on the Penn Inventory of Scrupulosity (PIOS; Abramowitz et al., 2002) which is the only psychometrically valid measure of scrupulosity available (Abramowitz and Jocoby, 2014). According to this approach, scrupulosity is represented by a continuum of severity of symptoms. Clinically relevant levels of scrupulosity could be represented by mean score in the PIOS higher than 1.42 (Huppert and Fradkin, 2016). This dimensional approach suited our approach which focus was on the measurement of scrupulosity in non-clinical sample. Thus, lower levels of fear of God and fear of sin could be regarded as an expression of religiosity. However, the high intensity of these experiences suggest the clinically relevant scruples. In the revised version of the manuscript we included explanation of this operationalization and of the approach used to measure scrupulosity. We also presented a violin boxplot which show how many participants reported a level of scrupulosity which was above the cut-off (36% of the participants). Thus, in the present study some participants reported clinically relevant levels of scrupulosity. Our analysis was focused on the associations between symptoms. Thus, we did not analyze the differences between these individuals and those who did not report the clinically relevant levels of scrupulosity. The cut-off used in this analysis also need further studies, therefore, we resign the analysis of differences.

lines 90-107 (in the manuscript with changes tracked): Two dimensions of scrupulosity identified in previous studies using the Penn Inventory of Scrupulosity (PIOS; Abramowitz et al., 2002) – the only psychometrically validated self-report measure of scrupulosity available to date (Abramowitz & Jacoby, 2014) – on clinical and non-clinical samples are: (a) the fear of having committed a religious or moral sin (Fear of sin), and (b) the fear of punishment from God (Fear of God; Abramowitz et al. 2002; Olatunji et al. 2007). Recent validation of the PIOS indicated that it consists of two factors which are: (a) fear of God and (b) fear of immorality (Hupper and Fradkin, 2016). Clinically relevant symptoms of scrupulosity captured by the PIOS include a strong fear of God’s punishment and of a lack of God’s acceptance, strong preoccupation by avoiding immoral thoughts, frequent fears of having immoral, sexual thoughts, a fear of acting immorally without being aware of it, and strong guilt (Huppert and Fradkin, 2016). In the current operationalization, scrupulosity is treated as a dimension ranging from less severe fears of being immoral to an extreme, obsessive fear of being immoral (Abramowitz et al., 2002). OCD patients suffering clinically relevant religious scruples reported higher levels of scrupulosity measured by the PIOS compared to individuals with other OCD-presentation and other diagnoses (e.g. anxiety disorders; Huppert and Fradkin, 2016; Siev et al., 2011).

lines 435-438 (in the manuscript with changes tracked): The mean total score of scrupulosity in the sample was lower than the cut-off for the clinically relevant level of scrupulosity (Huppert and Fradkin, 2016). However, a violin boxplot (Figure 1) shows that 36% of the participants reported scores higher than this cut-off.

Reviewer #1: In addition, the use of network analysis as an approach to examine the inter-connected patterns between variables are lack of argumentation. In line 64-65, the author(s) explained that network analysis posits that mental disorders could be better understood and theorized as the result of a causal interplay between symptoms in a network structure. However, I think that the use of network analysis as an analytical approach should be explained more broadly in terms of its advantages in understanding the patterns compared to different analytical tools (e.g., structural modelling, etc.). I have tried to find the answers for a question in my mind on why network analysis is used by carefully reading point 1.5 in the manuscript, but I am still thinking more argumentation is needed. 

The Authors: Thank you for this suggestion. In the revised version of the manuscript we elaborated on the importance of using the network analysis in studying scrupulosity.

lines 234-251 (in the manuscript with changes tracked): The network analysis helps also to investigate the centrality of symptoms and their communities, not necessarily assuming that latent processes are responsible for the detected associations between symptoms (Borsboom and Cramer, 2013).

In the study of scrupulosity, this approach seems particularity important. Mental health professionals, but also patients, have problems with differentiating between the normative religious struggle and pathological symptoms of scrupulosity. Religious doubts and rituals may appear similar to compulsive rituals and obsessions present in a mental disorder (e.g. OCD; Siev et al., 2021). It is also understudied how religious in-dividuals become scrupulous (Siev et al., 2016). People who are highly religious could develop scruples due to their rigid religiosity (Hendrson et al., 2022) but also due to their particular self-reference which lacks self-compassion and is overly perfectionistic (Brodar et al., 2015). Investigating the associations between symptoms of normative religious experiences, scrupulosity, OCD symptoms, and self-reference could help in determining which symptoms are responsible for the potential transformation of nor-mative religious doubts into clinically relevant obsessions, e.g. by determining which of them are bridge symptoms between normative processes and clinically-relevant symptoms (see a similar approach to parental burnout in Blanchard and Heeren, 2021).

Reviewer #1: In the method section, especially related to the explanation of the measures used for the data collection, I am thinking that the religiosity measure needs to be explained in more detail, especially related to where this scale came from.

The Authors: We included more in-depth description of the items used to measure religiosity and noted its limitations.

lines 371-381 (in the manuscript with changes tracked): Religiosity was measured with three items: “I identify strongly with my religious beliefs”, “My religion is an important part of my identity” and “How frequently do you attend religious ceremonies?”. The first two items were assessed on the scale ranged from 1 (Completely disagree) to 5 (Completely agree). They were based on the items used by Siev et al. (2011) which concern the importance of religious beliefs and the role of religion in one’s identity. The last item was assessed on the scale from 0 (Never) to 4 (Very frequent). The last item is frequently used as an indicator of religiosity (Abramowitz et al., 2002; Fincham and May, 2019). These three items have high internal consistency (α = .902) and were indicators of a single latent variable (χ2 = 0; RMSEA = 0; SRMR < .001; CFI = 1.00; GFI = 1.00).

lines 690-698 (in the manuscript with changes tracked): Third, the measurement of religiosity was limited. We used only three items to measure identification with religion, the role of religion for one’s identity, and the attendance of religious ceremonies. These items were very similar to the measurement of numerous studies on the association between religiosity and scrupulosity (Siev et al., 2011) and between religiosity and other psychological processes, e.g. forgiveness (Fincham and May, 2021). However, a more in-depth analysis of various dimensions of religiosity (e.g. centrality of religiosity) could allow for a better investigation into the associations between the role of religion for an individual and scruples.

Reviewer #1: Overall, I think that the nature of more exploratory (rather than confirmatory) analysis in this study needs to be acknowledged. 

The Authors: We believe that the exploratory analysis in the manuscript is justified by the fact that the scrupulosity is an understudied phenomenon (Siev et al., 2021). As we noted above, both clinicians and patients tend to confuse the symptoms of normative religious rituals or struggles and scrupulosity. Moreover, these associations are also understudied in non-clinical population. In such a population, when scrupulosity is not a manifestation of OCD, the localization of scruples in the context of religious struggles and subclinical OCD symptoms could be particularly important for the correct understanding of religious doubts which could develop to OCD symptoms (Sive et al., 2021). Moreover, to our knowledge this is one of the first studies which investigated the associations between religious struggle and scrupulosity in non-clinical sample. It was another reason to exploratory study the associations between religious/spiritual struggle and scrupulosity. We included this explanations in the revised manuscript.

lines 72-78 (in the manuscript with changes tracked): Since scrupulosity is understudied in its position in the context of normative religious experiences and psychopathological symptoms (Siev et al., 2021), we used the network analysis to exploratory analyze the position of scrupulosity in a network consisting of OCD symptomatology, religious and spiritual struggle, self-compassion and religiosity. We also compared this approach with the convenient approach based on latent variables in order to detect advantages and limitation of each approach in studying scrupulosity.

lines 263-272 (in the manuscript with changes tracked): Since scrupulosity was not frequently examined in the context of dynamic associations between religious struggles, psychopathological symptoms and self-compassion, we used the network analysis in the present study. This approach helped to detect central symptoms of the network and bridge symptoms which could be responsible for the activation of other symptoms in the network, and to extract the communities of symptoms (namely, clusters of symptoms which are strongly interconnected but less correlated with other symptoms present in the network. However, we also used the conventional structural equation modeling (SEM) in order to examine the associations between self-compassion, lack of self-compassion, religiosity, religious struggles, OCD symptoms, and scrupulosity.

Reviewer 2 Report

The idea of the paper is really interesting, with a deeper expression of aims and including the way to control strange/mediator/modulator variables. I have several comments about the article:

 Abstract

The abstract is well descripted, with information about the methods that is not included in Procedure part (line 14)

Introduction

Authors say “Religion was identified as one of the most common themes of obsessions (McKay et 24 al 2004) with an approximately 6% prevalence among patients suffering from OCD in a 25 field trial (Foa et al. 1995)". (lines 23-26). Both references are antique and probably out of context, I highly recommend to justify the study with actual references. 

Methods

It´s necessary more information about the design of the study. Directly it starts with Sample and settings, with no longer explanation. 

Results

I don´t know if the authors have included in the study the use of other statistics more than descriptive and correlation ones. It would be much more interesting with more analysis.Other thing is to consider a study about Religiosity with only 3 items about Religion. anyway, data and results are very well presented.

Discussion.

Authors include in the discussion aspects very interesting but not very extended. In my opinion, it´s very interesting than religiosity did not correlate with scrupulosity, but it´s necessary to clarify how and the implications for psychologists and religious researchers. .

Limitations of the study

There is a non-written limitation about the sample: neither mention about strange variables and possible influences in data, especially with a social media collected sample (fundamental in this kind of studies)

Author Response

Response to the Reviewer#2

Dear Reviewer,

Thank you for all insightful suggestions regarding the previous version of the manuscript. Below, you will find our responses to all your suggestions.

Thank you very much for your effort put in the correction of the previous version of our manuscript. We hope that changes that we made address correctly your suggestions.

The Authors

Reviewer #2: The idea of the paper is really interesting, with a deeper expression of aims and including the way to control strange/mediator/modulator variables. I have several comments about the article:

Abstract

The abstract is well descripted, with information about the methods that is not included in Procedure part (line 14)

The Authors: We corrected the abstract to precisely reflect the measures used in the study (lines 12-15):

lines 12-15 (in the manuscript with changes tracked): We applied the Self-Compassion Scale, Religious and Spiritual Struggle Scale, Obsessive-Compulsive Inventory-Revised, Pennsylvania Inventory of Scrupulosity, and questions concerning identification with religious beliefs, the role of religion in one’s identity and religious attendance.

Reviewer #2: Introduction

Authors say “Religion was identified as one of the most common themes of obsessions (McKay et 24 al 2004) with an approximately 6% prevalence among patients suffering from OCD in a 25 field trial (Foa et al. 1995)". (lines 23-26). Both references are antique and probably out of context, I highly recommend to justify the study with actual references. 

The Authors: We corrected this sentence and introduce more up-to-date references. The corrected sentence is:

lines 25-28 (in the manuscript with changes tracked): Religion was identified as one of the most common themes of obsessions (McKay et al 2004) with prevalence among patients suffering from OCD ranging from 0% to even 93% (Foa et al. 1995; Greenberg and Huppert, 2010; Huppert and Fradkin, 2016; Siev et al., 2021).

Reviewer #2: Methods

It´s necessary more information about the design of the study. Directly it starts with Sample and settings, with no longer explanation. 

The Authors: We included the following sentences in order to inform about the study design:

lines 274-280 (in the manuscript with changes tracked): The present study employs a cross-sectional design based on the quantitative measurement of religious struggles, scrupulosity, self-compassion, and OCD symptoms, controlling for a basic aspects of religiosity. This approach allows a reliable and valid assessment of the intensity of scrupulosity and its associations with other variables. Thus, we use the Pennsylvania Inventory of Scrupulosity which is currently the most significant measure of assessment of religious scrupulosity particularly suitable in discriminating scrupulous obsessions in Christians (Huppert and Fradkin, 2016).

Reviewer #2: Results

I don´t know if the authors have included in the study the use of other statistics more than descriptive and correlation ones. It would be much more interesting with more analysis. Other thing is to consider a study about Religiosity with only 3 items about Religion. anyway, data and results are very well presented.

The Authors: In the present study we were focused on the associations between scrupulosity and various symptoms of religious struggles, OCD symptomatology and self-compassion. Thus, our primary goal was to analyze the network of these symptoms. Thus, we used the network analysis which is appropriate to examine the patterns of relationships between variables, analyzing their structure, interactions and roles they play in the given network (Borsboom and Cramer, 2013; Trahair et al., 2020). This approach is based on the Gaussian graphical model (GGM; Costantini et al., 2015) operationg on the pairwise correlations between variables of interest in order to create a network. The GGM did not necessarily imply the existence of latent variables underlying the associations between observed variables. The network analysis includes a number of algorithms which helps in determining the centrality of examined variables in the network and to extract communities of the variables. In our study we used regularization of the netwok by the EBIClasso algorithm which shrinks the small partial correlations coefficients to zero and providing a basis for conditional independence of given variables. According to this procedure the graphical depiction is constructed. Next, we used bootstrapping methods to estimate the stability of the network and of the indices of the centrality of the symptoms. Next, we used spin glass algorithm to detect communities of symptoms and we investigated bridge influence in order to detect bridge symptoms which are responsible for the association between groups of symptoms (Epskamp et al., 2018). Thus, although our analysis was based on partial correlations between variables, a number of statistical methods were used to determine: (a) the central symptoms in the network; (b) the bridge symptoms between group of symptoms; (c) stability of the network (how stable is the structure of association between symptoms); (d) whether symptoms group into communities which are consistent with the theoretical approach or not (e.g. we demonstrated that scrupulosity in non-clinical sample is in the same community as religious/spiritual struggle).

            Moreover, according to the Reviewer suggestion, we also investigated the examined association using SEM. The results were similar to those obtained using the network analysis. In the SEM approach also, scrupulosity appeared to be correlated with religious/spiritual struggle and OCD symptoms (lines 554-577).

Epskamp, S., Borsboom, D. & Fried, E.I. Estimating psychological networks and their accuracy: A tutorial paper. Behav Res 50, 195–212 (2018). https://doi.org/10.3758/s13428-017-0862-1

We are aware of the limitation regarding the measure of religiosity used in the present study. Thus, we clearly stated that this measurement is a limitation of the present study (lines. 690-698). However, in other studies on scrupulosity (e.g. Abramowitz et al., 2002) of forgiveness (Fincham and May, 2019) religiosity was measured with even lower number of items (e.g. 2 or 3). In the present study we address two aspects of religiosity which were also measured in these studies, namely: religious attendance and personal importance of religion. We used one item measuring religious attendance (as in Fincham and May, 2019) and two addressing how religious beliefs are important for a participant and how a religious affiliation/religion is important for his/her identity. These items were constructed with reference to the 6-item measure of religiosity used by Siev and colleagues (2016). We used only three items because recruitment procedure included invitation for individuals who described themselves as religious person which was equivalent of items regarding religiosity/spirituality of an individual. Two questions about following the requirements of the religion we summarized in one item: I identify strongly with my religious beliefs, and next two items regarding the role of religiosity in one’s identity we also summarized into one item: “My religion is an important part of my identity”. We believe that these measurement was appropriate for the purposes of the present study which was more focused on the place of scrupulosity in religious struggles.

Reviewer #2: Discussion.

Authors include in the discussion aspects very interesting but not very extended. In my opinion, it´s very interesting than religiosity did not correlate with scrupulosity, but it´s necessary to clarify how and the implications for psychologists and religious researchers. .

The Authors: The discussion was substantially developed. In the revised version we discuss obtained results in more detail (lines 589-666). We also compare results obtained using network analysis and SEM, and we draw some suggestions for future studies concerning combined statistical analysis using this two techniques (lines 667-683).

lines 589-666 (in the manuscript with changes tracked): The community examination using the spin glass algorithm in the network analysis demonstrated that scrupulosity was a part of the same symptom community as reli-gious/spiritual struggles. However, SEM indicated that this association was not a result of a common latent process. Thus, religious struggles and scrupulosity seem to activate each other in the network, but are not the reflections of the same psychological reli-gious process. Religious scruples could be correlated with religious doubts and strug-gles, but scrupulosity is not the next type of religious struggles. This finding is im-portant from the clinical perspective. Most scrupulous individuals perceive their symptoms as interfering with their religious experience (Siev et al., 2011). Thus, clini-cians should help their clients to differentiate between normative religious doubts and scruples. It is also important to avoid normalization of religious scruples by religious communities (Abramowitz and Buchholtz, 2020). Future studies should also investi-gate the direction of associations between scrupulosity and religious struggles using longitudinal designs. This finding is in line with recent studies indicating that religious crisis, but not fundamentalism, was correlated with scrupulosity (Henderson et al., 2022).

Scrupulosity was also found as a correlation of low self-compassion which is con-sistent with previous studies (Borgogna et al., 2020; Fisak et al., 2019). This result indi-cates that a lack of self-kindness and poor insight could foster scrupulosity among reli-gious people (Tolin et al., 2001). Contrarily, isolation, self-judgement and over-identification could also create a ground for a rigid, lacking self-forgiveness ap-proach to one’s sin (Brodar et al. 2015). SEM demonstrated that scrupulosity was fre-quent among individuals with lower self-compassion, but the association was weak. Similarly, network analyses indicated a relatively weak association between self-compassion and scrupulosity. These results suggest that treatment of scrupulosity could benefit from introducing self-compassion exercises, but focusing only on the de-velopment of the self-compassionate attitude seems to be not enough to treat scruples.

Contrary to expectations, religiosity did not correlate with scrupulosity (see Abramowitz and Buchholtz, 2020). Although a number of studies demonstrated posi-tive associations between religiosity and religious scruples (Siev et al., 2021), some previous studies did not show significant associations between these variables (Nelson et al., 2006). The associations between religiosity and scrupulosity, therefore, seem to be complex and could depend on the particular cultural context (Abramowitz and Jacoby, 2014). Also, contrary to the previous findings concerning associations between religiosity and obsessiveness (Inozu et al., 2012), religiosity correlated negatively with OCD symptoms. However, previous findings demonstrated that religious fundamen-talism positively correlated with compulsiveness and obsessiveness (Inozu et al., 2012). In the present study, we did not measure such a dimension of religiosity as fundamen-talism. Thus, only particularly rigid and extreme religiosity could foster OCD symp-tomatology, while other dimensions of religiosity could be differently related to OCD symptoms. Although the correlational and network analysis in the present study did not demonstrate significant association between religiosity and scrupulosity, SEM showed a positive yet weak association. This result could be due to the sampling pro-cedure, in which we invited the participants who describe themselves as religioussug-gest that the latent variable reflecting religiosity is associated with a higher probability of religious scruples. However, this finding could also be a result of a negative covaria-tion of religiosity and OCD symptoms, which were positively associated with scrupu-losity. Religiosity could play some role in scruples, but scruples were not a simple con-sequence of religiosity itself (Abraowitz and Buchholtz, 2020). However, the findings indicate that high identification or high importance of religion for one’s identity are not risk factors for scrupulosity. Future studies should use a more precise measure of religiosity to examine these associations in more detail. From the practical point of view, our findings indicate that religious scruples could appear among people who did not perceive themselves as particularly religious in terms of religious attendance and the role of religiosity in their identity. A fear of hell or condemnation could be a serious reason of psychological distress and compulsions also among non-religious individu-als.

The present study showed that scrupulosity is positioned on the intersection be-tween the mental experiences associated with tension, strain, and conflicts about sa-cred matters (Exline et al. 2014) and obsessing reflecting OCD symptomatology. The bridge analysis indicated that demonic, moral and religious doubt struggles, obsessing, and scrupulosity were all positively correlated. In the network analysis approach, this may indicate that there could be an internal dynamic between these symptoms, and one symptom could activate others. Similarly, SEM demonstrated that religious strug-gles and OCD symptoms were positively associated and both predicted higher scrupu-losity. From the clinical point of view, it could indicate that strong religious struggles may activate scrupulosity which could turn into obsessing. This could illustrate a pro-cess of developing pathologized religiosity out of religious struggles experienced by in-dividuals without proper insight and a healthy attitude toward the self. The reverse pattern is also possible, namely when OCD-related obsessions “invade” a sphere of re-ligiosity. Although such causal interpretations of the current results is unjustified, fu-ture studies should investigate the possibility of causal relationships between these symptoms in longitudinal approach. The present study, however, showed that reli-gious scruples seem to be more closely linked to religious struggles than OCD symp-toms. This finding could encourage clinicians to pay more attention in conversation with their clients to foster their insight into boundaries of normative religious struggles and to develop their knowledge about characteristics of religious scruples. These find-ings also indicated that patients with proneness to develop obsessive symptomatology (e.g., due to personality factors such as difficulty with change or maladaptive perfec-tionism; Fang et al., 2016; Siev et al., 2021) could be at a higher risk of developing reli-gious scruples also due to their stronger religious struggles.

lines 667-683 (in the manuscript with changes tracked): Both the network analysis approach and the SEM approach yielded similar results concerning scrupulosity. When focusing on the associations between symptoms or la-tent variables, scrupulosity appeared as a phenomenon at the intersection between normative religious struggle and OCD symptomatology. The network analysis demon-strated that the central among these symptoms are Fear of sin and Obsessing. Thus, these symptoms should be the targets of treatment of individuals suffering from reli-gious scruples. This result also demonstrated the advantage of the network analysis which allows us to detect the particular symptoms which could activate strongly other symptoms in the studied network. A combination of both statistical methods indicted that although religious struggles and scrupulosity belong to one community of symp-toms, the processes underlying these symptoms are different. Thus, scruples are proba-bly not simply religious doubts and should not be disregarded in the examination of clients’ religiosity. From the methodological point of view, future studies on psycho-pathology of scruples could benefit from the combined methodology of the network analysis and the latent structures analysis. Future studies should examine more pre-cisely the role of religiosity in the studies network. SEM indicated that religiosity could be associated with higher scrupulosity, while the network analysis indicated that re-ligiosity and scruples were relatively independent.

Reviewer #2: Limitations of the study

There is a non-written limitation about the sample: neither mention about strange variables and possible influences in data, especially with a social media collected sample (fundamental in this kind of studies)

The Authors: Thank you for this suggestion. In the revised version of the manuscript we included the following sentences regarding this limitation.

lines 685-690 (in the manuscript with changes tracked): Second, the participants were not representative of the population of religious individuals in Poland. Although recruitment through social media is suitable for populations that are difficult to reach and could help participants in maintaining anonymity, this sampling method is not without limitations (Topolevec-Vranic and Natarajan, 2016). Thus, future studies should investigate the associations between studied variables in representative samples and in direct contact.

Round 2

Reviewer 2 Report

I really appreciate the efforts of the authors for including all the comments and suggestions, in my opinion the article is completed.